# DATA, NOT MODEL: EXPLAINING BIAS TOWARD LLM TEXTS IN NEURAL RETRIEVERS

## ABSTRACT

Recent studies show that neural retrievers often display source bias, favoring passages generated by LLMs over human-written ones, even when both are semantically similar. This bias has been considered an inherent flaw of retrievers, raising concerns about the fairness and reliability of modern information access systems. Our work challenges this view by showing that source bias stems from supervision in retrieval datasets rather than the models themselves. We found that non-semantic differences, like fluency and term specificity, exist between positive and negative documents, mirroring differences between LLM and human texts. In the embedding space, the bias direction from negatives to positives aligns with the direction from human-written to LLM-generated texts. We theoretically show that retrievers inevitably absorb the artifact imbalances in the training data during contrastive learning, which leads to their preferences over LLM texts. To mitigate the effect, we propose two approaches: 1) reducing artifact differences in training data and 2) adjusting LLM text vectors by removing their projection on the bias vector. Both methods substantially reduce source bias. We hope our study alleviates some concerns regarding LLM-generated texts in information access systems.

## 1 INTRODUCTION

The rapid rise of large language models (LLMs) has reshaped the information landscape, creating corpora where human-written and LLM-generated texts coexist. Within this hybrid ecosystem, an emerging phenomenon has been observed: neural retrievers often prefer LLM-generated passages over semantically similar human-written ones, a phenomenon known as source bias (Dai et al., 2024b;c). This bias raises concerns at multiple levels. For users, it risks diminishing search quality by ranking fluent but less relevant or even misleading LLM outputs above more relevant human-authored content. For human creators, it undermines fairness by systematically downranking their work and reducing its visibility. At the ecosystem level, it may amplify LLM-generated text through self-reinforcing feedback loops, further marginalizing human contributions (Chen et al., 2024; Zhou et al., 2024).

Given these significant concerns, understanding the root cause of source bias is crucial. Prior work offers different explanations: Dai et al. (2024b) attribute the bias to architectural similarities between retrievers built on pretrained language models (PLMs) and LLMs, while Wang et al. (2025) argue that retrievers prefer low-perplexity texts, a property often exhibited by LLM outputs. However, it remains unclear why such preferences emerge, and no explanation has been widely accepted. Consequently, recent efforts have shifted toward mitigating source bias, for example, through causal debiasing to reduce the impact of perplexity (Wang et al., 2025) or by aligning LLM outputs to be less biased for retrievers (Dai et al., 2025).

In this paper, we aim to uncover the root cause of source bias in neural retrievers. Specifically, we address three research questions (RQs):

- **RQ1: Is source bias a general property of neural retrievers?** Beyond the commonly studied retrievers trained on MS MARCO (Nguyen et al., 2016), we examine two additional families: (1) general-purpose embedding models trained for diverse tasks such as clustering, classification, semantic similarity, and retrieval, and (2) unsupervised retrievers trained without relevance annotations, such as Contriever (Izacard et al., 2021) and SimCSE (Gao et al., 2021). We find that these models exhibit only mild source bias, whereas fine-tuning the unsupervised retrievers on MS

MARCO induces severe bias. This suggests that source bias is not inherent to neural retrievers but is largely introduced through relevance supervision.

- **RQ2: Why does relevance supervision induce source bias?** Our analysis of 14 retrieval datasets uncovers systematic non-semantic differences between positive and negative documents, including variations in fluency, as measured by perplexity, and lexical specificity. These differences closely mirror the distinctions between LLM-generated and human-authored texts. In the embedding space, we further observe that the bias direction from negatives to positives aligns strongly with the direction from human-written to LLM-generated texts. Theoretical analysis confirms that retrievers trained with contrastive losses inevitably absorb these imbalances.
- **RQ3: How can source bias be mitigated?** We propose two mitigation strategies: (1) reducing artifact differences in training data to prevent retrievers from encoding non-semantic factors, and (2) debiasing embeddings by subtracting the projection of LLM-generated vectors on the bias direction. Both approaches substantially reduce source bias, confirming that it originates from systematic imbalances in relevance annotations.

In summary, we challenge the prevailing view that neural retrievers are inherently biased toward LLM-generated texts. Instead, we show that source bias arises from artifact imbalances in retrieval datasets rather than model architecture. Our findings highlight two complementary pathways for mitigation: curating training data to minimize non-semantic artifacts and explicitly decoupling artifact effects in retrievers. With a deeper understanding of source bias, LLM-generated texts need not be regarded as inherently problematic. We hope this study alleviates concerns about their use and fosters a more objective perspective on integrating LLM-generated data into retrieval systems.

## 2 RELATED WORK

**Source Bias in Information Retrieval.**   Dai et al. (2024c) revealed that neural retrievers exhibit a clear preference for LLM-generated passages even when their semantic content is similar to human-written ones, a phenomenon termed *source bias*. Cocktail (Dai et al., 2024a) further established a benchmark to evaluate this phenomenon across diverse retrieval datasets systematically. Similar effects have also been noted in related IR scenarios, including multimodal retrieval (Xu et al., 2024), recommender systems (Zhou et al., 2024), and retrieval-augmented generation (Chen et al., 2024), underscoring the view that source bias is a broad challenge in the LLM era.

**Mechanisms and Mitigation.**   Prior work has examined both explanations and mitigations for source bias. Early studies linked it to architectural similarity between PLMs and LLMs (Dai et al., 2024c). Wang et al. (2025) showed that PLM-based retrievers overrate low-perplexity documents, and Dai et al. (2024b) framed the issue more broadly as a distribution mismatch. Mitigation approaches include retriever-side methods such as causal debiasing (Wang et al., 2025) and LLM-side methods like LLM-SBM (Dai et al., 2025). Following these perspectives, prior work has often assumed that source bias is a universal property of neural retrievers. By contrast, we evaluate a broader spectrum of retrievers and show that source bias is not inherent to neural retrievers. We further develop a retriever-centric theory and conduct a set of experiments indicating that the bias largely arises from supervision, and we provide practical mitigations.

## 3 RQ1: IS SOURCE BIAS A GENERAL PROPERTY OF NEURAL RETRIEVERS

The previously discussed phenomenon of source bias (Dai et al., 2024b;c) has been mainly observed in retrieval-supervised models, which are trained on relevance-labeled datasets such as MS MARCO (Nguyen et al., 2016). This observation prompts us to examine whether source bias is a general property of neural retrievers or a phenomenon largely induced by relevance supervision.

We therefore design two controlled experiments to disentangle the role of supervision from model architecture: (1) we examine whether source bias persists in models beyond those primarily fine-tuned on retrieval datasets, considering both general-purpose embedding models and unsupervised retrievers; and (2) we assess the impact of retrieval supervision by fine-tuning several unsupervised retrievers on MS MARCO while holding architecture fixed. Next, we present the model families, datasets, and metrics used in these experiments.

Table 1: $\Delta$NDSR@5 results across 14 datasets for 13 neural retrievers spanning three model families. Negative values are shaded in red to indicate a preference for LLM-generated passages, while positive values are shaded in blue to indicate a preference for human-written passages. Asterisks (*) denote statistically significant deviations from zero (two-sided t-test, $p < 0.05$).

| Dataset (↓) | Relevance-Supervised Retrievers | | | | | General-Purpose Embedding Models | | | | | Unsupervised Retrievers | | |
|---|---|---|---|---|---|---|---|---|---|---|---|---|---|
| | ANCE | TAS-B | coCondenser | RetroMAE | DRAGON | BGE | BCE | GTE | E5 | M3E | Contriever | E5-Unsup | SimCSE |
| MS MARCO | -0.040* | -0.119* | -0.018* | -0.080* | -0.081* | -0.021* | 0.084* | -0.074* | -0.036* | 0.053* | 0.280* | 0.094* | 0.384* |
| DL19 | -0.073 | -0.224* | -0.072 | -0.180* | -0.233* | -0.017 | 0.119 | -0.178* | 0.015 | 0.139 | 0.271* | 0.086 | 0.428* |
| DL20 | -0.029 | -0.070 | -0.078 | -0.081 | -0.116* | 0.057 | 0.048 | -0.049 | 0.012 | 0.203* | 0.275* | 0.190* | 0.389* |
| NQ | -0.040* | -0.074* | -0.067* | -0.055* | -0.096* | -0.078* | 0.324* | -0.003 | 0.153* | 0.040* | 0.186* | 0.228* | 0.140* |
| NFCorpus | -0.087* | -0.082* | -0.067* | -0.098* | -0.079* | 0.030 | -0.064* | -0.142* | 0.034 | -0.143* | -0.083* | -0.348* | 0.127* |
| TREC-COVID | -0.162* | -0.328* | -0.340* | -0.193* | -0.133* | 0.014 | -0.025 | -0.236* | -0.118 | -0.085 | -0.135* | -0.224* | 0.162* |
| HotpotQA | -0.015* | -0.011* | -0.008* | -0.013* | 0.014* | 0.061* | 0.184* | 0.010* | 0.078* | 0.063* | -0.273* | -0.091* | 0.097* |
| FiQA-2018 | -0.179* | -0.169* | -0.257* | -0.244* | -0.160* | -0.150* | 0.414* | -0.050* | -0.116* | 0.102* | -0.068* | -0.052* | 0.210* |
| Touché-2020 | -0.101 | -0.165* | -0.128* | -0.099 | -0.052* | -0.042 | 0.218* | -0.017 | -0.185* | 0.242* | -0.133* | -0.073 | 0.027 |
| DBpedia | -0.095* | -0.039* | -0.053* | -0.077* | -0.054* | 0.017 | 0.069* | -0.035* | 0.003 | 0.019 | -0.130* | -0.062* | 0.064* |
| SCIDOCS | -0.040* | -0.054* | -0.058* | -0.073* | -0.048* | -0.061* | 0.517* | -0.046* | 0.010 | 0.275* | 0.028* | 0.059* | 0.268* |
| FEVER | -0.199* | -0.024* | -0.032* | -0.006* | -0.040* | 0.040* | 0.306* | -0.027* | 0.031* | 0.031* | 0.028* | -0.008* | 0.031* |
| Climate-FEVER | -0.314* | -0.082* | -0.153* | -0.105* | -0.091* | -0.038* | 0.642* | -0.080* | 0.215* | 0.123* | -0.003 | 0.017 | 0.070* |
| SciFact | -0.024 | -0.058* | -0.049* | -0.048* | -0.041* | 0.011 | 0.015 | -0.079* | 0.004 | -0.206* | 0.017 | -0.101* | -0.059* |

## 3.1 EXPERIMENTAL SETUP

**Model Families.** We evaluate three distinct families of models: (A) *Relevance-Supervised Retrievers*, trained with direct or distilled supervision signals derived from large-scale human relevance annotations (e.g., MS MARCO), including ANCE(Xiong et al., 2020), TAS-B (Hofstätter et al., 2021), coCondenser (Gao & Callan, 2021), RetroMAE (Xiao et al., 2022), and DRAGON (Lin et al., 2023); (B) *General-Purpose Embedding Models*, trained on large and diverse corpora with multi-task objectives beyond retrieval (e.g., semantic textual similarity, clustering, and classification) and widely adopted in Retrieval-Augmented Generation (RAG) applications, including BGE (Xiao et al., 2023), BCE (NetEase Youdao, 2023), GTE (Li et al., 2023), E5 (Wang et al., 2022), and M3E (Wang Yuxin, 2023); (C) *Unsupervised Retrievers*, trained without any human relevance annotations, typically via self-supervised contrastive objectives, including Contriever (Izacard et al., 2021), unsupervised Sim-CSE (Gao et al., 2021), and the unsupervised variant of E5 (Wang et al., 2022).

**Datasets.** Following recent work on source bias (Wang et al., 2025; Dai et al., 2025), We conduct experiments on the Cocktail benchmark (Dai et al., 2024a), which pairs human-written passages with LLM-generated counterparts that are semantically similar. In particular, we use the 14 datasets in Cocktail that originate from BEIR (Thakur et al., 2021), covering diverse domains such as open-domain QA, scientific retrieval, fact verification, and argumentative search. All datasets and model checkpoints are from publicly available HuggingFace releases to ensure reproducibility, with links and dataset statistics reported in Appendix B and Appendix C.

**Preference Metrics.** Prior work has shown that relevance-based metrics can conflate retrieval quality with source preference. To isolate preference from relevance, Huang et al. (2025) proposed the Normalized Discounted Source Ratio (NDSR), which measures the proportion of retrieved documents from a given source type within the top-$k$ results:

$$\text{NDSR}_c@k = \frac{\sum_{i=1}^{k} \mathbb{1}\left(\text{source}(d_i) = c\right) \cdot w_i}{\sum_{i=1}^{k} w_i}, \qquad \Delta\text{NDSR}@k = \text{NDSR}_{\text{Human}}@k - \text{NDSR}_{\text{LLM}}@k.$$

Here, $c \in \{\text{Human}, \text{LLM}\}$ specifies the source category being measured; $\mathbb{1}(\cdot)$ is an indicator that returns 1 when the document $d$ at rank $i$ originates from source $c$ and 0 otherwise; $w_i = 1/\log_2(1+i)$ is a rank discount that assigns higher weight to higher-ranked positions; and $k$ denotes the evaluation depth, i.e., the top-$k$ retrieved documents. We use $\Delta\text{NDSR}@k$ as our main preference metric, which ranges from $-1$ to $1$: positive values indicate a preference for human-written passages, while negative values indicate a preference for LLM-generated passages.

## 3.2 EXPERIMENTAL RESULTS

Having established the model families, datasets, and evaluation metrics, we now turn to the results of our two controlled experiments. These experiments separate the influence of retrieval supervision from differences across retriever families.

**Source Bias across Retriever Families.** We first examine whether source bias extends beyond Relevance-Supervised Retrievers to other model families. Table 1 presents $\Delta$NDSR@5 results on 14 datasets for all three families. The results show that *Relevance-Supervised Retrievers* consistently favor LLM-generated passages, with negative scores on nearly all datasets, aligning with prior observations of source bias in this category. In contrast, *General-Purpose Embedding Models* and *Unsupervised Retrievers* show no consistent pattern, with preferences varying across datasets in both directions. This suggests that source bias is not consistently present across all retriever families. In addition to these source-preference results, we also report the retrieval effectiveness of all models in Appendix D for completeness.

**Impact of Supervision on Source Bias.** We then turn to the second experiment, where we fine-tune unsupervised retrievers on MS MARCO. In their base form (Table 1), Contriever, E5-Unsup, and Sim-CSE display only mild or inconsistent source preferences. After fine-tuning, however, all three models exhibit a clear shift toward favoring LLM-generated passages, as shown in Table 2. This contrast indicates that retrieval supervision is a key factor driving the observed source bias.

**Role of Passage Length.** A potential contributing factor in the above analysis is passage length. Neural retrievers are known to exhibit non-semantic length biases, often assigning disproportionately high scores to shorter passages (Thakur et al., 2024; Fayyaz et al., 2025). Meanwhile, the LLM-generated passages in the Cocktail benchmark are typically shorter than their human-written counterparts, raising the question of whether the source preference observed in supervised retrievers merely reflects a preference for shorter text.

Table 2: $\Delta$NDSR@5 results of unsupervised retrievers after MS MARCO fine-tuning, corresponding to the same base models in Table 1. The "-FT" suffix denotes fine-tuning on MS MARCO. Negative values are shaded in red to indicate a preference for LLM-generated passages, while positive values are shaded in blue to indicate a preference for human-written passages. Asterisks (*) denote statistically significant deviations from zero (two-sided t-test, $p < 0.05$).

| Dataset ($\downarrow$) | Relevance-Supervised Retrievers | | |
|---|---|---|---|
| | Contriever-FT | E5-FT | SimCSE-FT |
| MS MARCO | 0.012* | -0.044* | -0.053* |
| DL19 | -0.035 | -0.198* | -0.133 |
| DL20 | 0.121* | 0.022 | -0.178* |
| NQ | -0.038* | -0.051* | -0.060* |
| NFCorpus | -0.139* | -0.189* | -0.060* |
| TREC-COVID | -0.282* | -0.271* | -0.205* |
| HotpotQA | -0.004 | -0.019* | -0.013* |
| FiQA-2018 | -0.215* | -0.212* | -0.189* |
| Touché-2020 | -0.087* | -0.196* | -0.169* |
| DBpedia | -0.010 | -0.036* | -0.053* |
| SCIDOCS | -0.050* | -0.072* | -0.041* |
| FEVER | -0.018* | -0.064* | 0.000 |
| Climate-FEVER | -0.099* | -0.091* | -0.049* |
| SciFact | -0.086* | -0.077* | -0.044* |

To assess the role of length, we construct a length-controlled variant of each dataset. In this version, the LLM-generated passages preserve the original semantics but are systematically lengthened. We then repeat the source-preference evaluation on this controlled setting. As detailed in Appendix J, relevance-supervised retrievers still prefer LLM-generated passages even when the LLM versions are longer than the human ones, although the strength of this preference becomes weaker. This indicates that passage length modulates the magnitude of source bias but does not explain its direction: supervised models continue to favor LLM-generated text even when length advantages are removed.

**Summary.** Taken together, these findings indicate that source bias is not an inherent property of neural retrievers but is largely induced by retrieval dataset supervision, motivating the next section on why relevance supervision gives rise to such bias.

## 4 RQ2: WHY DOES RELEVANCE SUPERVISION INDUCE SOURCE BIAS?

Since source bias is largely induced by relevance supervision, we now examine why such supervision leads retrievers to prefer LLM-generated text. We hypothesize that supervised datasets introduce systematic imbalances in non-semantic artifacts between positive and negative passages, such as fluency and lexical specificity. These imbalances lead retrievers to learn to exploit these stylistic cues alongside semantic content. Positive passages in retrieval datasets are often polished and information-dense to resemble high-quality answers, a stylistic pattern that coincides with LLM-generated text. This overlap explains why retrievers tend to favor LLM-generated passages during inference. We examine this mechanism through linguistic analyses, embedding-space evidence, and a theoretical decomposition of the retrieval objective.

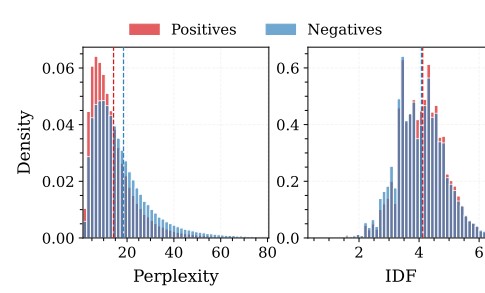 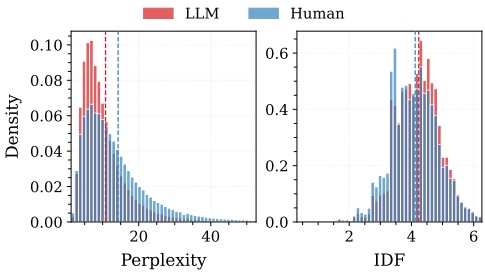

(a) Positives vs. Negatives.

(b) LLM-generated vs. Human-written.

Figure 1: Distribution of perplexity and inverse document frequency. (a) Comparison between annotated positives and the negatives in training supervision. (b) Comparison between LLM-generated and human-written passages. In both settings, the first group (Positives / LLM) exhibits lower PPL and higher IDF, revealing parallel artifact imbalances. Dashed lines indicate means.

## 4.1 LINGUISTIC ANALYSES

To examine whether positive passages and LLM-generated passages share similar stylistic patterns, we conduct linguistic analyses. We focus on two complementary features: perplexity (PPL), which captures fluency, and inverse document frequency (IDF), which captures lexical specificity.

*Perplexity (PPL).* Given a passage $d = (w_1, \ldots, w_{|d|})$ with $|d|$ tokens, its perplexity under a language model $p_\theta$ is defined as $\text{PPL}(d) = \exp\left(-\frac{1}{|d|} \sum_{i=1}^{|d|} \log p_\theta(w_i \mid w_{<i})\right)$. Lower PPL corresponds to more predictable and fluent text under the model. We compute PPL using Llama-3-8B-Instruct(Dubey et al., 2024), a strong open-weight model whose broad training distribution makes it a useful automatic fluency measure commonly adopted in recent LLM-based evaluation pipelines.

*Inverse Document Frequency (IDF).* For a token $t$, its IDF is defined as $\text{IDF}(t) = \log \frac{N}{1+\text{df}(t)}$, where $N$ is the total number of documents in the corpus and $\text{df}(t)$ is the number of documents containing $t$. Passage-level IDF is computed as the median of token-level IDF values within the passage, which provides robustness to outliers. We estimate IDF statistics on the full MS MARCO collection ($\sim$8.8M passages), using the standard tokenizer from the Apache Lucene library for passage segmentation (Hatcher & Gospodnetic, 2004).

**Training Data: Positives vs. Negatives.** We begin by examining the artifact imbalance between positives and negatives in training data, using MS MARCO as a representative case. Specifically, we define the positive pool as the union of passages annotated as relevant to at least one training query, and the negative pool as all remaining passages. While the negative pool contains sparse false negatives, the majority are non-relevant, making it a representative sample for linguistic analysis.

Figure 1a shows that positives have lower perplexity (PPL) and a slight increase in inverse document frequency (IDF) compared to the negatives. Both differences are statistically significant; the difference in PPL is larger, while the effect of IDF is statistically reliable but small(see Appendix F for detailed statistics). Overall, positives are more fluent and marginally higher lexical specificity. This pattern is linguistically natural: annotated positives are often drawn from the main content of edited sources (e.g., news articles, Wikipedia entries, product pages), whereas the negatives covers a wider range of raw web text (e.g., forums, boilerplate, semi-structured fragments) that typically introduce disfluencies and lexically less specific patterns.

Taken together, these findings show that relevance-labeled datasets exhibit artifact imbalance, as exemplified by MS MARCO. Beyond MS MARCO, we also observe consistent PPL imbalances across other IR datasets (Appendix F), suggesting that this tendency is a general property of retrieval supervision rather than an idiosyncrasy of a single dataset. This raises the question of whether similar imbalances also arise when contrasting passages by source.

**Source Type: LLM-generated vs. Human-written Passages.** To investigate this question, we compare LLM-generated passages with their human-written counterparts on the 14 BEIR-derived datasets from the Cocktail benchmark. For clarity of presentation, Figure 1b reports representative

results on MS MARCO, where LLM-generated passages exhibit lower PPL and higher IDF than human passages, with statistically significant differences of moderate effect size(see Appendix F for detailed statistics). This pattern aligns with how LLMs are trained: pretraining on large, relatively curated corpora encourages more formal and information-dense language, yielding outputs that are more polished and lexically informative. Complete results across all 14 datasets are provided in Appendix F, with consistent patterns observed across all datasets.

**Summary.** Taken together, the analyses show that the artifact imbalances between positives and negatives are consistent with those between LLM-generated and human-written passages. This consistency suggests that source bias may arise from the same underlying stylistic imbalances shared between supervised datasets and LLM-generated text.

While perplexity and IDF serve as illustrative examples, they do not capture the full spectrum of stylistic artifacts. To move beyond linguistic features and connect more directly to the mechanisms of neural retrieval, we next examine how such imbalances are encoded in the embedding space.

## 4.2 EMBEDDING-SPACE SHIFTS

In this section, we investigate whether the embedding shift induced by supervision (positives vs. negatives) aligns with the shift induced by source type (LLM-generated vs. human-written passages). To address this, we proceed in three steps: (1) estimate the direction separating positives from negatives; (2) estimate the direction separating LLM-generated from human-written passages and assess its stability; and (3) evaluate whether the two directions are aligned.

**Notation.** Let $q$ denote a query and $d$ denote a passage. For supervised retrieval, we write $d^+$ and $d^-$ for an annotated positive and a sampled negative passage; for source-type analysis, we write $d^{\text{LLM}}$ and $d^{\text{Human}}$ for an LLM-generated passage and its human-written counterpart. The query and document encoders $h_q(\cdot)$ and $h_d(\cdot)$ map $q$ and $d$ to embeddings in $\mathbb{R}^m$, where $m$ is the embedding dimension, and the retrieval score is given by $s_\theta(q, d) = \langle h_q(q), h_d(d) \rangle$.

We use $\delta$ to denote a displacement vector between paired embeddings, such as the LLM–Human displacement $\delta^{\text{LH}} = h_d(d^{\text{LLM}}) - h_d(d^{\text{Human}})$. The symbol $\bar{\delta}$ denotes the average displacement over a set of paired passages (e.g., across a dataset). $\mathbb{E}[\cdot]$ denotes expectation over the indicated distribution.

**Estimating the Positive–Negative Embedding Direction.** To estimate an embedding direction that primarily reflects stylistic artifacts rather than semantic variation, it is important to ensure that the positive and negative pools have comparable semantic distributions. In MS MARCO, however, positives and negatives differ systematically in topical coverage. Following common practice in (Karpukhin et al., 2020), we mitigate this by retrieving the top-10 BM25 candidates for each query and randomly sampling one as the negative, yielding a 1:1 pairing with the annotated positive. This construction balances topical distributions, allowing the mean embedding contrast between positives and negatives to more accurately isolate non-semantic artifacts. Formally, we estimate the supervision-induced positive–negative embedding direction as $\bar{\delta}_{\text{PN}} = \mathbb{E}\big[h_d(d^+) - h_d(d^-)\big]$.

**Significance Criterion in High-Dimensional Space.** Before turning to the LLM–Human direction, we first establish a statistical threshold to test whether displacement vectors exhibit a coherent direction rather than random noise. In 768 dimensions, random vectors are almost orthogonal, with cosine similarities concentrated around zero. Over 99.7% of random pairs fall within $\pm 3\sigma$ of the mean (Appendix G). Deviations beyond this range therefore indicate a consistent, non-random effect. We use this as the significance criterion for subsequent analyses.

**Is the LLM–Human Distinction a Stable Embedding Direction?** Unlike the positive–negative setting, the LLM–Human comparison uses semantically aligned counterparts, allowing us to directly compute pairwise displacements. For each aligned pair, we define

$$\delta_i^{\text{LH}} = h_d(d_i^{\text{LLM}}) - h_d(d_i^{\text{Human}}).$$

We then examine whether these displacements form a coherent embedding-space direction, evaluating their stability across three complementary dimensions of consistency.

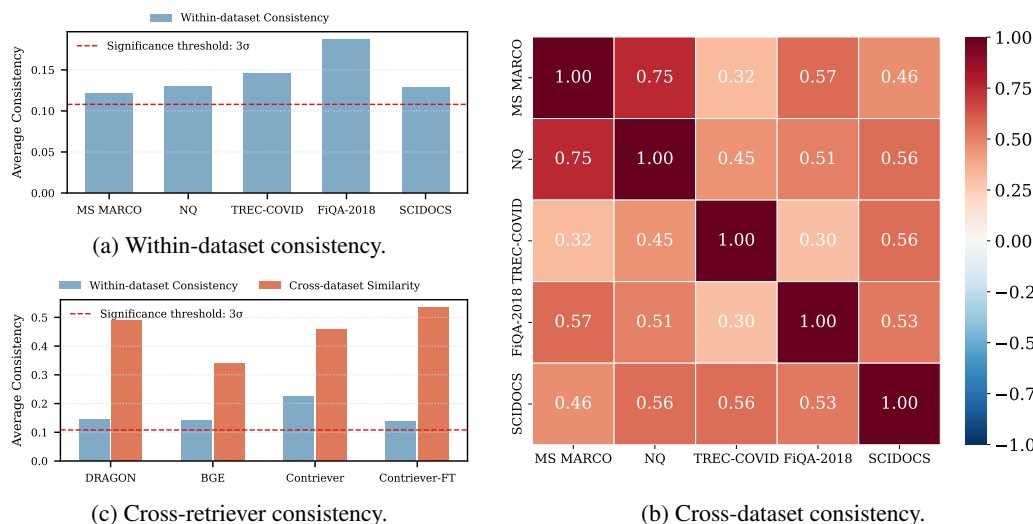

(a) Within-dataset consistency.

(c) Cross-retriever consistency.

(b) Cross-dataset consistency.

Figure 2: The LLM–Human distinction forms a stable embedding-space direction. The plots demonstrate this consistency along three dimensions: (a) within datasets, (b) across datasets, and (c) across retrievers. All metrics shown exceeded the $3\sigma$ threshold. Plots (a, b) use the DRAGON retriever; results for all 14 datasets are in Appendix H.

**(1) Within datasets.** We test whether displacement vectors exhibit mutual alignment by computing the average pairwise cosine similarity $\mathbb{E}_{i \neq j}[\cos(\delta_i^{LH}, \delta_j^{LH})]$. Values exceeding the $3\sigma$ significance threshold indicate a consistent, non-random shift within each dataset (Figure 2a).

**(2) Across datasets.** For each dataset $D$, we compute the dataset-level mean displacement $\overline{\delta}_{LH,D} = \mathbb{E}_{d_i \in D}[\delta_i^{LH}]$, and evaluate cross-dataset alignment via $\cos(\overline{\delta}_{LH,D_1}, \overline{\delta}_{LH,D_2})$, which tests whether datasets share the same underlying direction (Figure 2b).

**(3) Across models.** As shown in Figure 2c, repeating the analysis with multiple retrievers shows that the LLM–Human displacement remains coherent both within and across datasets, and consistent across all retrievers examined.

Together, these findings demonstrate that the LLM–Human distinction reflects a stable embedding direction shared across datasets and models, rather than an artifact of any specific retriever or dataset.

**Do the Positive–Negative and LLM–Human Directions Align?** Having established that the LLM–Human distinction corresponds to a stable embedding direction, we now test our central hypothesis: whether this direction aligns with the supervision-induced positive–negative direction, $\overline{\delta}_{PN}$. We measure this alignment by computing the cosine similarity between the mean LLM–Human direction for each dataset, $\overline{\delta}_{LH,D}$, and the positive–negative direction derived from MS MARCO. As shown in Figure 3a, the alignment is consistently strong and statistically significant across all datasets. Furthermore, this effect is not specific to a single retriever. Figure 3b shows that the alignment remains robustly significant across retrievers. This strong, consistent alignment demonstrates that the positive-negative and LLM-human distinctions correspond to a shared direction in the embedding space. We now turn to our theoretical framework to formalize the mechanism by which this alignment emerges as a learnable shortcut for relevance, thus inducing source bias.

### 4.3 THEORETICAL FRAMEWORK: ARTIFACT ENCODING IN NEURAL RETRIEVERS

Building on the linguistic and embedding-space analyses, we formalize these observations in a theoretical framework. For clarity and intuition, this section presents an informal overview of our key results (see Appendix E for formal statements and proofs). Our theory shows that (1) whenever training data contains systematic artifact imbalances, the retriever necessarily learns these non-semantic cues, and (2) these cues manifest as an approximately linear component in the retrieval score.

To illustrate this, we abstractly decompose any document $d$ into its semantic features $M_d$ and its non-semantic artifact features $A_d$ (e.g., fluency, lexical patterns). An *artifact imbalance* exists if positive

passages systematically differ from negative passages in their artifact features. Specifically, we define the artifact imbalance at training time as the difference between the expected artifact features of positive and negative documents: $\Delta_A = \mathbb{E}[A_{d+}] - \mathbb{E}[A_{d-}]$. Here $A_{d+}$ and $A_{d-}$ represent the artifact features of positive and negative documents, respectively.

Our first key result is that such imbalance directly shapes the optimal retriever's scoring function.

**Proposition 1** (Decomposition of the Optimal Scorer, Informal)**.** *The Bayes-optimal retrieval score $s^*(\cdot, \cdot)$, which is approximated by models trained with contrastive objectives like InfoNCE, necessarily decomposes into a semantic term and an artifact-dependent term:*

$$s^*(q, d) = \text{Score}_{\text{semantic}}(q, M_d) + \text{Score}_{\text{artifact}}(q, A_d).$$

*If the training data exhibit artifact imbalance ($\Delta_A \neq 0$), the artifact-dependent term is non-zero.*

> **Insight 1:** Artifact imbalance forces the optimal retriever to encode non-semantic cues. The model learns that artifacts like high fluency are predictive of relevance, creating a shortcut.

Next, we connect this decomposition to the practical implementation of dot-product retrievers.

**Proposition 2** (Embedding-Space Decomposition, Informal)**.** *For a standard dot-product retriever, the retrieval score $s_\theta(\cdot, \cdot)$ can be approximated as a sum of a semantic and an artifact-based score:*

$$s_\theta(q, d) = \langle h_q(q), h_d(d) \rangle \approx \underbrace{\langle h_q(q), h_d^{\text{sem}}(d) \rangle}_{\text{semantic}} + \underbrace{\langle h_q(q), h_d^{\text{art}}(d) \rangle}_{\text{artifact}}.$$

This decomposition can be viewed as a first-order Taylor approximation. The document encoder, though a complex non-linear model, can be locally approximated as linear in the artifact features, which is consistent with our empirical observation of a stable direction in embedding space.

> **Insight 2:** The artifact-based score is captured by a linear operation in the embedding space.

**Why Other Families Do Not Exhibit Consistent Source Bias.** Unlike relevance-supervised retrievers, other retriever families do not exhibit a consistent source bias. (1) General-purpose embedding models are trained on diverse tasks such as semantic textual similarity, natural language inference, clustering, and classification. Many of these objectives are symmetric: if sentence $a$ is a positive for sentence $b$, then $b$ is a positive for $a$. Such symmetry prevents systematic differences between "positives" and "negatives," yielding $\Delta_A \approx 0$ and avoiding artifact-driven shortcuts. (2) Unsupervised retrievers like Contriever rely on self-supervised objectives constructed directly from raw corpora, where adjacent spans of text are treated as positives and other in-batch samples serve as negatives. Because no annotated positive–negative splits are involved, the training signal lacks systematic stylistic imbalance. In both cases, the artifact-dependent term in Proposition 1 averages out in expectation, explaining why these models do not exhibit a consistent source bias (Section 3).

**Summary.** Our analyses consistently show that source bias arises from artifact imbalance in training data. Linguistically, positives in supervision and LLM-generated passages both show lower perplexity and increased lexical specificity than their counterparts. In embedding space, the supervision-induced positive–negative direction and the LLM–human displacement align as a stable, shared axis. Our theoretical framework formalizes this observation: any artifact imbalance in training necessarily introduces a linear artifact component into the retriever's scoring function. This explains why stylistic imbalances observed in supervision manifest as a stable embedding direction spuriously aligned with relevance, providing both a mechanistic account of source bias and a foundation for mitigation strategies.

## 5 RQ3: HOW CAN SOURCE BIAS BE MITIGATED?

Building on our theoretical results, we now move from explanation to mechanism validation and bias mitigation. Proposition 1 revealed that artifact imbalance ($\Delta_A \neq 0$) in supervision necessarily leads the retriever to encode non-semantic cues, while Proposition 2 showed that these cues manifest as a linear component in embedding space. These insights suggest two complementary strategies:

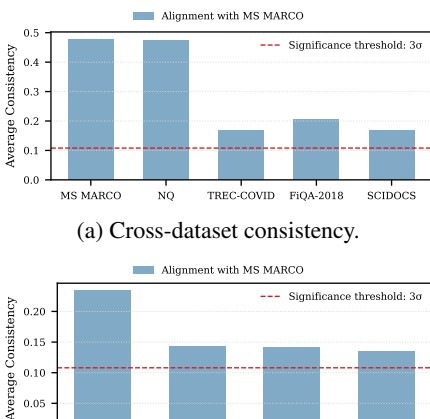

(a) Cross-dataset consistency.

(b) Cross-retriever consistency.

Figure 3: The LLM–Human displacement aligns with the positive–negative supervision direction. Panel (a) shows cross-dataset consistency, and panel (b) shows cross-retriever consistency. Across both settings, cosine similarities exceed the $3\sigma$ threshold, confirming a stable and coherent embedding-space direction.

Table 3: $\Delta$NDSR@5 results under different negative sampling strategies. "In-batch only" suppresses artifact imbalance ($\Delta_A \approx 0$), "Standard" combines in-batch and hard negatives, and "Hard-neg only" maximizes artifact imbalance. Shading in the Average row (with the color bar on the right) indicates the relative magnitude of $|\Delta$NDSR@5$|$, with darker colors representing stronger source bias relative to the "Hard-neg only" configuration.

| | In-batch only | Standard | Hard-neg only | % |
|---|---|---|---|---|
| MS MARCO | 0.014 | -0.051 | -0.057 | 0 |
| DL19 | 0.025 | -0.155 | -0.182 | |
| DL20 | 0.041 | -0.120 | -0.152 | 10 |
| NQ | 0.020 | -0.081 | -0.085 | 20 |
| NFCorpus | -0.050 | -0.068 | -0.093 | 30 |
| TREC-COVID | -0.182 | -0.252 | -0.285 | 40 |
| HotpotQA | 0.003 | 0.017 | -0.021 | |
| FiQA-2018 | -0.055 | -0.227 | -0.238 | 50 |
| Touché-2020 | -0.077 | -0.202 | -0.193 | 60 |
| DBPedia | -0.021 | -0.041 | -0.043 | 70 |
| SCIDOCS | 0.010 | -0.051 | -0.035 | |
| FEVER | 0.014 | -0.005 | -0.013 | 80 |
| Climate-FEVER | -0.032 | -0.071 | -0.080 | 90 |
| SciFact | -0.032 | -0.051 | -0.053 | |
| **Average** | **-0.024** | **-0.099** | **-0.109** | 100 |

reduce $\Delta_A$ during training or suppress the artifact direction at inference. Importantly, these interventions not only mitigate source bias but also validate its underlying mechanism: if reducing $\Delta_A$ or removing the artifact direction reliably diminishes bias, this provides strong empirical support for our theoretical account. In summary, our aim is not to advance state-of-the-art debiasing, but to substantiate the mechanism of source bias and propose simple interventions that are readily applicable in practice. We therefore examine both strategies below.

**Training-time Interventions: Controlling Artifact Imbalance ($\Delta_A$).** We propose a simple training-time mitigation strategy: adopting *in-batch only* negative sampling, where negatives are exclusively other queries' positives from the annotated pool. This setup ensures $\mathbb{E}[A_{d+}] \approx \mathbb{E}[A_{d-}]$ and thus suppresses artifact imbalance ($\Delta_A \approx 0$). To evaluate its effectiveness, we contrast it against two reference settings: (1) the *standard* sampling scheme widely used for training neural retrievers, which combines in-batch negatives with one mined hard negative per query and yields a moderate $\Delta_A$; and (2) a *hard-neg only* setting, which draws negatives solely from the unannotated pool and maximizes $\Delta_A$. Together, these three conditions provide a controlled spectrum of artifact imbalance.

For fairness and controllability, we fine-tune BERT-based retrievers on MS MARCO using the official BEIR pipeline (Devlin et al., 2019; Thakur et al., 2021), modifying only the negative sampling strategy while keeping all other factors fixed. This isolates the impact of sampling on source bias.

As shown in Table 3, the in-batch only strategy substantially reduces source bias, improving the average $\Delta$NDSR@5 from -0.099 (standard sampling) to -0.024, whereas standard and hard-neg only sampling lead to progressively stronger bias. Although omitting mined hard negatives slightly impairs retrieval effectiveness (average NDCG@5 drops from 0.493 to 0.475, see Appendix I), the reduction in bias is considerable. These findings validate our theoretical account and demonstrate that mitigation at training time is indeed effective, providing a useful pivot for further exploration of debiasing strategies. Building on this, we next examine inference-time interventions that suppress artifact directions without retraining.

**Inference-time Interventions: Suppressing Artifact Directions.** Our analyses in Section 4.2 showed that LLM-generated passages induce a consistent displacement in embedding space. Let $n = \frac{\bar{\delta}_{LH}}{\|\bar{\delta}_{LH}\|}$ denote the normalized mean displacement between LLM rewrites and their human counterparts. In practice, we estimate $n$ by averaging displacement vectors from 1000 randomly sampled human–LLM passage pairs per dataset. This sampling size yields stable estimates across datasets

Table 4: $\Delta$NDSR@5 results (original vs. debiased) across 5 datasets and 5 relevance-supervised retrievers. Positive values indicate a preference for human-written passages, whereas negative values indicate a preference for LLM-generated ones. In the Average row, the first line reports the mean $\Delta$NDSR@5, and the second line shows the remaining proportion of $|\Delta$NDSR@5$|$ after debiasing (original = 100%). Shading in the Average row reflects the relative magnitude of $|\Delta$NDSR@5$|$, with darker colors indicating stronger source bias. Full results on all 14 datasets appear in Appendix I.

| Dataset ($\downarrow$) | ANCE | | coCondenser | | DRAGON | | RetroMAE | | TAS-B | | % |
|---|---|---|---|---|---|---|---|---|---|---|---|
| | Original | Debias | Original | Debias | Original | Debias | Original | Debias | Original | Debias | 0 |
| MS MARCO | -0.042 | 0.168 | -0.020 | 0.094 | -0.083 | -0.065 | -0.083 | 0.011 | -0.121 | -0.062 | |
| TREC-COVID | -0.162 | -0.178 | -0.340 | -0.281 | -0.134 | -0.154 | -0.194 | -0.098 | -0.328 | -0.248 | 20 |
| NQ | -0.042 | -0.032 | -0.072 | -0.071 | -0.099 | -0.085 | -0.060 | -0.044 | -0.078 | -0.062 | 40 |
| FiQA-2018 | -0.179 | -0.159 | -0.219 | -0.263 | -0.161 | -0.154 | -0.205 | -0.201 | -0.170 | -0.182 | |
| SCIDOCS | -0.040 | 0.069 | -0.058 | -0.053 | -0.048 | -0.012 | -0.073 | 0.007 | -0.054 | 0.010 | 60 |
| **Average** | -0.093 | -0.026 | -0.142 | -0.115 | -0.105 | -0.094 | -0.123 | -0.072 | -0.150 | -0.109 | 80 |
| | (100%) | (28%) | (100%) | (81%) | (100%) | (90%) | (100%) | (59%) | (100%) | (73%) | 100 |

Table 5: NDCG@5 results (original vs. debias) on 5 datasets for 5 relevance-supervised retrievers. Full results on 14 datasets are provided in Appendix I.

| Dataset ($\downarrow$) | ANCE | | coCondenser | | DRAGON | | RetroMAE | | TAS-B | |
|---|---|---|---|---|---|---|---|---|---|---|
| | Original | Debias | Original | Debias | Original | Debias | Original | Debias | Original | Debias |
| MS MARCO | 0.590 | 0.568 | 0.620 | 0.621 | 0.665 | 0.665 | 0.626 | 0.626 | 0.617 | 0.617 |
| TREC-COVID | 0.679 | 0.690 | 0.707 | 0.695 | 0.684 | 0.681 | 0.744 | 0.737 | 0.644 | 0.638 |
| NQ | 0.628 | 0.626 | 0.687 | 0.687 | 0.737 | 0.737 | 0.704 | 0.704 | 0.689 | 0.689 |
| FiQA-2018 | 0.255 | 0.255 | 0.244 | 0.244 | 0.323 | 0.322 | 0.278 | 0.277 | 0.257 | 0.261 |
| SCIDOCS | 0.114 | 0.113 | 0.124 | 0.125 | 0.148 | 0.146 | 0.136 | 0.136 | 0.138 | 0.133 |
| **Average** | 0.453 | 0.450 | 0.477 | 0.474 | 0.511 | 0.510 | 0.497 | 0.496 | 0.468 | 0.467 |

while remaining computationally efficient. At inference, for passage embedding $v \in \mathbb{R}^m$ (i.e., $v = h_d(d)$), we suppress the component along $n$: $v' = v - \langle v, n \rangle n$.

We focus on five relevance-supervised retrievers, where source bias is most pronounced and our theoretical analysis directly applies. As shown in Tables 4 and 5, the projection reduces source bias in most cases, while retrieval effectiveness is largely preserved. Importantly, it requires no retraining and adds negligible computational cost, as embeddings are already computed during inference. This provides a practical drop-in solution that can be readily integrated into existing retrieval systems.

**Summary.** These interventions jointly achieve mechanism validation and mitigation. Training-time sampling strategies directly manipulate $\Delta_A$, showing a consistent trend where larger imbalance leads to stronger bias, thereby establishing a clear link between supervision artifacts and source bias. Inference-time projection complements this by suppressing artifact-driven directions in embedding space, reducing bias with negligible cost and no retraining. Together, these complementary approaches both reinforce our theoretical account and provide practical strategies for mitigating source bias in deployed retrieval systems.

## 6 CONCLUSION

This paper re-examines the origins of source bias in neural retrieval and shows that it is not an inherent property but a learned consequence of artifact imbalance in supervised training data. Through theoretical analysis and empirical validation, we demonstrate how contrastive objectives encode non-semantic artifacts and how LLM-generated text mirrors these artifacts, producing a consistent biased direction in embedding space. Building on this insight, we introduce two mitigation methods: (1) a training-time negative sampling control that effectively mitigates source bias, and (2) an inference-time projection that achieves similar debiasing strength while largely preserving retrieval performance. Our findings indicate that artifact imbalance is an important factor behind source bias, motivating the development of de-artifacted datasets and training practices for more robust and fair retrieval systems. More broadly, the analyses and mitigation strategies explored here may inform the study of other spurious correlations across domains.

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

## A  THE USE OF LARGE LANGUAGE MODELS (LLMS)

In this study, we employed Large Language Models (LLMs) as an AI writing assistant, using them strictly to improve the clarity and readability of our textual expressions. The models were not used for research ideation, literature retrieval, or discovery, nor to generate any substantive suggestions.

## B  REPRODUCIBILITY RESOURCES

To ensure reproducibility, we provide the full list of datasets and model checkpoints used in this work. All datasets and models are obtained from publicly available HuggingFace releases or their official websites. Our usage strictly follows the respective licenses and research-only terms of the original sources. Tables 6 and 7 provide direct links for reference.

Table 6: Datasets used in this paper (Cocktail versions) and their HuggingFace links.

| Dataset | HuggingFace Link |
|---|---|
| MS MARCO (Nguyen et al., 2016) | https://huggingface.co/datasets/IR-Cocktail/msmarco |
| TREC-DL'19 (Craswell et al., 2020) | https://huggingface.co/datasets/IR-Cocktail/dl19 |
| TREC-DL'20 (Craswell et al., 2021) | https://huggingface.co/datasets/IR-Cocktail/dl20 |
| Natural Questions (Kwiatkowski et al., 2019) | https://huggingface.co/datasets/IR-Cocktail/nq |
| NFCorpus (Boteva et al., 2016) | https://huggingface.co/datasets/IR-Cocktail/nfcorpus |
| TREC-COVID (Voorhees et al., 2021) | https://huggingface.co/datasets/IR-Cocktail/trec-covid |
| HotpotQA (Yang et al., 2018) | https://huggingface.co/datasets/IR-Cocktail/hotpotqa |
| FiQA-2018 (Maia et al., 2018) | https://huggingface.co/datasets/IR-Cocktail/fiqa |
| Touché-2020 (Bondarenko et al., 2020) | https://huggingface.co/datasets/IR-Cocktail/webis-touche2020 |
| DBpedia-Entity (Hasibi et al., 2017) | https://huggingface.co/datasets/IR-Cocktail/dbpedia-entity |
| SCIDOCS (Cohan et al., 2020) | https://huggingface.co/datasets/IR-Cocktail/scidocs |
| FEVER (Thorne et al., 2018) | https://huggingface.co/datasets/IR-Cocktail/fever |
| Climate-FEVER (Diggelmann et al., 2020) | https://huggingface.co/datasets/IR-Cocktail/climate-fever |
| SciFact (Wadden et al., 2020) | https://huggingface.co/datasets/IR-Cocktail/scifact |

Table 7: Dense retriever checkpoints used in this paper and their HuggingFace links.

| Model | HuggingFace Link |
|---|---|
| *Relevance-Supervised Retrievers* | |
| ANCE (Xiong et al., 2020) | https://huggingface.co/sentence-transformers/msmarco-roberta-base-ance-firstp |
| TAS-B (Hofstätter et al., 2021) | https://huggingface.co/sentence-transformers/msmarco-distilbert-base-tas-b |
| coCondenser (Gao & Callan, 2021) | https://huggingface.co/sentence-transformers/msmarco-bert-co-condensor |
| RetroMAE (Xiao et al., 2022) | https://huggingface.co/nthakur/RetroMAE_BEIR |
| DRAGON (query encoder) (Lin et al., 2023) | https://huggingface.co/nthakur/dragon-plus-query-encoder |
| DRAGON (corpus encoder) (Lin et al., 2023) | https://huggingface.co/nthakur/dragon-plus-context-encoder |
| *General-Purpose Embedding Models* | |
| BGE-base (Xiao et al., 2023) | https://huggingface.co/BAAI/bge-base-en-v1.5 |
| BCE (NetEase Youdao, 2023) | https://huggingface.co/maidalun1020/bce-embedding-base_v1 |
| GTE (Li et al., 2023) | https://huggingface.co/thenlper/gte-base |
| E5 (Wang et al., 2022) | https://huggingface.co/intfloat/e5-base-v2 |
| M3E (Wang Yuxin, 2023) | https://huggingface.co/moka-ai/m3e-base |
| *Unsupervised Retrievers* | |
| Contriever (Izacard et al., 2021) | https://huggingface.co/nishimoto/contriever-sentencetransformer |
| E5-Unsupervised (Wang et al., 2022) | https://huggingface.co/intfloat/e5-base-unsupervised |
| SimCSE (Gao et al., 2021) | https://huggingface.co/princeton-nlp/unsup-simcse-bert-base-uncased |

## C  DATASET STATISTICS

Table 8 summarizes the statistics of the 14 datasets used in this paper. This table is adapted from the Cocktail benchmark (Dai et al., 2024a), with minor modifications.

## D  RETRIEVAL EFFECTIVENESS OF EVALUATED MODELS

For completeness, we report the retrieval effectiveness of all evaluated models on the Cocktail benchmark. Table 9 presents NDCG@5 across 14 datasets for the 13 retrievers spanning the three model families. Table 10 further reports results after fine-tuning unsupervised retrievers on MS MARCO. These results complement the source preference analyses in Section 3.

## E  FORMAL STATEMENTS AND PROOFS

We formalize the intuition that artifact imbalance biases retrieval by analyzing how it affects the retriever's learning objective in three steps: (1) derive the Bayes-optimal retrieval scorer, (2) decompose it into semantic and artifact terms, and (3) relate this decomposition to an embedding-space view that bridges theory with practical retriever representations.

**Notation and Setting.** Let $q$ denote a query and $d$ a document. Each document $d$ is associated with semantic features $M_d$ and artifact features $A_d$ (e.g., perplexity, IDF profile, stylistic attributes), both treated as random vectors. We consider dense retrievers consisting of a dual-encoder and a scoring function. The dual-encoder maps queries and documents into embeddings $h_q(q), h_d(d) \in \mathbb{R}^m$, and a typical scoring function is the inner product $s_\theta(q, d) = \langle h_q(q), h_d(d) \rangle$.

Training relies on positive and negative query–document pairs. Let $p_{\text{pos}}(q, d)$ denote the distribution of positive pairs, and let $p(q)p(d)$ be the reference distribution given by independent sampling of queries and documents. Positives $(q, d^+)$ are drawn from $p_{\text{pos}}(q, d)$, while negatives $(q, d^-)$ are sampled from $p(q)p(d)$—a standard abstraction of in-batch and hard-negative schemes. We define the *artifact imbalance* at training time as $\Delta_A = \mathbb{E}[A_{d^+}] - \mathbb{E}[A_{d^-}]$.

Table 8: Statistics of the 14 datasets in the Cocktail benchmark used in this paper. Avg. D/Q denotes the average number of relevant documents per query. This table is adapted from Dai et al. (2024a).

| Dataset | Domain | Task | Relevancy | #Pairs | #Queries | #Corpus | Avg. D/Q | Avg. Length (Q / Human / LLM) |
|---|---|---|---|---|---|---|---|---|
| MS MARCO | Misc. | Passage Retrieval | Binary | 532,663 | 6,979 | 542,203 | 1.1 | 6.0 / 58.1 / 55.1 |
| DL19 | Misc. | Passage Retrieval | Binary | - | 43 | 542,203 | 95.4 | 5.4 / 58.1 / 55.1 |
| DL20 | Misc. | Passage Retrieval | Binary | - | 54 | 542,203 | 66.8 | 6.0 / 58.1 / 55.1 |
| TREC-COVID | Biomedical | Biomedical IR | 3-level | - | 50 | 128,585 | 430.1 | 10.6 / 197.6 / 165.9 |
| NFCorpus | Biomedical | Biomedical IR | 3-level | 110,575 | 323 | 3,633 | 38.2 | 3.3 / 221.0 / 206.7 |
| NQ | Wikipedia | QA | Binary | - | 3,446 | 104,194 | 1.2 | 9.2 / 86.9 / 81.0 |
| HotpotQA | Wikipedia | QA | Binary | 169,963 | 7,405 | 111,107 | 2.0 | 17.7 / 67.9 / 66.6 |
| FiQA-2018 | Finance | QA | Binary | 14,045 | 648 | 57,450 | 2.6 | 10.8 / 133.2 / 107.8 |
| Touché-2020 | Misc. | Argument Retrieval | 3-level | - | 49 | 101,922 | 18.4 | 6.6 / 165.4 / 134.4 |
| DBpedia | Wikipedia | Entity Retrieval | 3-level | - | 400 | 145,037 | 37.3 | 5.4 / 53.1 / 54.0 |
| SCIDOCS | Scientific | Citation Prediction | Binary | - | 1,000 | 25,259 | 4.7 | 9.4 / 169.7 / 161.8 |
| FEVER | Wikipedia | Fact Checking | Binary | 140,079 | 6,666 | 114,529 | 1.2 | 8.1 / 113.4 / 91.1 |
| Climate-FEVER | Wikipedia | Fact Checking | Binary | - | 1,535 | 101,339 | 3.0 | 20.2 / 99.4 / 81.3 |
| SciFact | Scientific | Fact Checking | Binary | 919 | 300 | 5,183 | 1.1 | 12.4 / 201.8 / 192.7 |

Table 9: NDCG@5 results across 14 datasets for 13 dense retrievers. Higher is better.

| Dataset ($\downarrow$) | Relevance-Supervised Retrievers | | | | | General-Purpose Embedding Models | | | | | Unsupervised Retrievers | | |
|---|---|---|---|---|---|---|---|---|---|---|---|---|---|
| | ANCE | TAS-B | coCondenser | RetroMAE | DRAGON | BGE | BCE | GTE | E5 | M3E | Contriever | E5-Unsup | SimCSE |
| MS MARCO | 0.647 | 0.680 | 0.683 | 0.688 | 0.735 | 0.688 | 0.590 | 0.688 | 0.702 | 0.473 | 0.504 | 0.575 | 0.245 |
| DL19 | 0.686 | 0.760 | 0.734 | 0.743 | 0.771 | 0.755 | 0.708 | 0.750 | 0.747 | 0.507 | 0.515 | 0.624 | 0.346 |
| DL20 | 0.701 | 0.724 | 0.708 | 0.751 | 0.758 | 0.729 | 0.651 | 0.718 | 0.743 | 0.489 | 0.492 | 0.597 | 0.289 |
| NQ | 0.640 | 0.708 | 0.711 | 0.746 | 0.790 | 0.778 | 0.625 | 0.789 | 0.790 | 0.494 | 0.623 | 0.737 | 0.353 |
| NFCorpus | 0.266 | 0.340 | 0.345 | 0.336 | 0.389 | 0.403 | 0.275 | 0.394 | 0.368 | 0.257 | 0.339 | 0.371 | 0.109 |
| TREC-COVID | 0.671 | 0.670 | 0.677 | 0.735 | 0.678 | 0.783 | 0.574 | 0.763 | 0.714 | 0.390 | 0.391 | 0.605 | 0.296 |
| HotpotQA | 0.553 | 0.705 | 0.663 | 0.747 | 0.799 | 0.792 | 0.533 | 0.761 | 0.801 | 0.575 | 0.650 | 0.668 | 0.369 |
| FiQA-2018 | 0.275 | 0.408 | 0.467 | 0.498 | 0.529 | 0.384 | 0.285 | 0.380 | 0.373 | 0.366 | 0.225 | 0.373 | 0.093 |
| Touché-2020 | 0.479 | 0.427 | 0.349 | 0.441 | 0.390 | 0.402 | 0.333 | 0.423 | 0.411 | 0.242 | 0.308 | 0.333 | 0.252 |
| DBpedia | 0.408 | 0.493 | 0.493 | 0.528 | 0.533 | 0.514 | 0.360 | 0.514 | 0.541 | 0.370 | 0.427 | 0.488 | 0.259 |
| SCIDOCS | 0.095 | 0.111 | 0.102 | 0.116 | 0.123 | 0.177 | 0.118 | 0.190 | 0.141 | 0.069 | 0.114 | 0.174 | 0.041 |
| FEVER | 0.820 | 0.835 | 0.842 | 0.870 | 0.876 | 0.928 | 0.682 | 0.924 | 0.905 | 0.865 | 0.878 | 0.925 | 0.510 |
| Climate-FEVER | 0.270 | 0.306 | 0.255 | 0.311 | 0.318 | 0.368 | 0.274 | 0.373 | 0.303 | 0.161 | 0.223 | 0.264 | 0.195 |
| SciFact | 0.465 | 0.602 | 0.549 | 0.611 | 0.631 | 0.715 | 0.533 | 0.732 | 0.688 | 0.448 | 0.614 | 0.719 | 0.239 |

**Step 1: Optimal scorer under InfoNCE.** InfoNCE is a widely used contrastive learning objective, which encourages the retriever to assign higher scores to positive pairs $(q, d^+)$ than to negatives $(q, d^-)$, thereby pulling queries closer to their relevant documents while pushing them away from irrelevant ones. The Bayes-optimal retriever is therefore given by the following lemma.

**Lemma 1.** *For contrastive learning with negatives sampled independently from $p(d)$, the Bayes-optimal scorer of a dense retriever is $s^*(q, d) = \log \frac{p_{\mathrm{pos}}(q,d)}{p(q)p(d)} + C$, where $C$ is an additive constant that does not depend on $d$.*

> **Insight 1:** Retriever training with InfoNCE is equivalent to estimating a log-density ratio.

**Step 2: Decomposition into semantic and artifact terms.** Building on this formulation, we view each document as consisting of semantic features $M_d$ and artifact features $A_d$, under which the density-ratio admits the following decomposition. In the main-text informal statement, these two terms are denoted $\mathrm{Score}_{\mathrm{semantic}}(q, M_d)$ and $\mathrm{Score}_{\mathrm{artifact}}(q, A_d)$. Here, $\phi(q, M_d)$ and $\psi(A_d \mid q, M_d)$ provide their formal counterparts.

**Proposition 3** (Formal version of Proposition 1). *Let $T(d) = (M_d, A_d)$ be a measurable mapping decomposing a document into semantic and artifact features. Then $\log \frac{p_{\mathrm{pos}}(q,d)}{p(q)p(d)} = \underbrace{\phi(q, M_d)}_{\text{semantic}} + \underbrace{\psi(A_d \mid q, M_d)}_{\text{artifact}}$. If the training sampler induces artifact imbalance (e.g., $\Delta_A \neq 0$), then the Bayes-optimal scorer necessarily carries an artifact-dependent term. In particular, $I(s^*(q, d); A_d \mid q, M_d) > 0$, where $I(\cdot; \cdot \mid \cdot)$ denotes conditional mutual information.*

> **Insight 2:** Whenever artifact imbalance exists, the Bayes-optimal scorer necessarily carries an artifact-dependent term.

**Step 3: An idealized embedding-space view.** To translate the above decomposition into an embedding-space view, we focus on the dot-product retriever. This corresponds to the informal

Table 10: NDCG@5 results of unsupervised retrievers after MS MARCO fine-tuning, corresponding to the same base models in Table 9. The "-FT" suffix denotes fine-tuning on MS MARCO.

| Dataset (↓) | Relevance-Supervised Retrievers | | |
| --- | --- | --- | --- |
| | Contriever-FT | E5-FT | SimCSE-FT |
| MS MARCO | 0.676 | 0.711 | 0.630 |
| DL19 | 0.696 | 0.763 | 0.727 |
| DL20 | 0.673 | 0.720 | 0.703 |
| NQ | 0.732 | 0.764 | 0.670 |
| NFCorpus | 0.339 | 0.378 | 0.279 |
| TREC-COVID | 0.446 | 0.731 | 0.590 |
| HotpotQA | 0.712 | 0.735 | 0.577 |
| FiQA-2018 | 0.255 | 0.336 | 0.220 |
| Touché-2020 | 0.347 | 0.428 | 0.389 |
| DBpedia | 0.495 | 0.532 | 0.444 |
| SCIDOCS | 0.117 | 0.138 | 0.083 |
| FEVER | 0.857 | 0.895 | 0.837 |
| Climate-FEVER | 0.289 | 0.312 | 0.261 |
| SciFact | 0.593 | 0.679 | 0.470 |

decomposition $h_d^{\text{sem}}(d)$ and $h_d^{\text{art}}(d)$ in the main text, with $h_{\text{sem}}(M_d)$ and $h_{\text{art}}(A_d)$ making the dependence on the underlying features explicit.

**Proposition 4** (Formal version of Proposition 2). *For a dot-product retriever with query encoder $h_q$ and passage encoder $h_d$, suppose each passage $d$ can be abstractly decomposed into semantic features $M_d$ and artifact features $A_d$. Then, under a linear approximation, $s_\theta(q, d) = \underbrace{\langle h_q(q), h_{\text{sem}}(M_d) \rangle}_{\text{semantic}} + \underbrace{\langle h_q(q), h_{\text{art}}(A_d) \rangle}_{\text{artifact (linear)}}$, where $h_{\text{sem}}(M_d)$ and $h_{\text{art}}(A_d)$ denote the semantic and artifact representations, respectively.*

> **Insight 3:** Under a linear approximation, the retriever's score explicitly decomposes into semantic and artifact contributions in the embedding space.

Formal proofs of Lemma 1, Proposition 3, and Proposition 4 are provided in Appendix E.1. Together, these results specify the conditions under which supervision can induce source bias: when training data exhibit artifact imbalance, the optimal scorer encodes artifact-dependent signals alongside semantic content. The analysis further predicts that such artifacts correspond to linearly decodable directions in the embedding space, offering a concrete signature for empirical validation. This perspective clarifies when and how source bias may emerge and provides testable predictions that motivate the empirical analyses that follow.

### E.1 PROOF OF LEMMA 1

This appendix provides the formal proofs of the main theoretical results presented in Section 4.3. Specifically, we include detailed proofs of Lemma 1, Proposition 3, and Proposition 4.

*Proof.* We derive the Bayes-optimal scorer for InfoNCE under independent negative sampling. The proof proceeds in three steps: (i) formalize the sampling and objective, (ii) show that risk minimization forces the predictor to match the true posterior, and (iii) compute this posterior and simplify.

**Step 1: Sampling scheme and objective.** Draw a query $q \sim p(q)$ and sample an index $I \sim \text{Unif}\{0, \ldots, K\}$, where $K$ is the number of negative samples (not to be confused with the evaluation depth $k$). Here $I$ denotes the index of the positive passage. We use the same symbol for mutual information $I(\cdot; \cdot)$ later, but the two usages are contextually disambiguated. Conditioned on $(q, I)$, sample the positive passage $d_I \sim p_{\text{pos}}(d \mid q)$ and sample negatives $d_j \sim p(d)$ for all $j \neq I$, yielding the candidate batch $\boldsymbol{d} = (d_0, \ldots, d_K)$.

Given scores $s(q, d_j) \in \mathbb{R}$, the model predicts

$$\pi_\theta(i \mid q, \boldsymbol{d}) = \frac{\exp\big(s(q, d_i)\big)}{\sum_{j=0}^{K} \exp\big(s(q, d_j)\big)}. \tag{1}$$

In practice, a temperature parameter $\tau$ is often included (i.e., $s(q,d) = \langle h_q(q), h_d(d) \rangle / \tau$). For clarity, we omit $\tau$, as it simply rescales the scores without affecting the derivation.

The InfoNCE loss is the expected negative log-likelihood (cross-entropy):

$$\mathcal{L}(\theta) = \mathbb{E}_{(q,\boldsymbol{d})}\Big[\mathbb{E}_{I|q,\boldsymbol{d}}\big[-\log \pi_\theta(I \mid q, \boldsymbol{d})\big]\Big] = \mathbb{E}_{(q,\boldsymbol{d})}\big[R(\boldsymbol{s}; q, \boldsymbol{d})\big], \tag{2}$$

where we denote $P_i = \mathbb{P}(I = i \mid q, \boldsymbol{d})$ and $\pi_i = \pi_\theta(i \mid q, \boldsymbol{d})$

$$R(\boldsymbol{s}; q, \boldsymbol{d}) = -\sum_{i=0}^{K} P_i \log \pi_i. \tag{3}$$

**Step 2: Bayes optimality.** This risk decomposes as

$$R(\boldsymbol{s}; q, \boldsymbol{d}) = -\sum_{i=0}^{K} P_i \log \pi_i = \underbrace{\Big(-\sum_i P_i \log P_i\Big)}_{H(P)} + \sum_i P_i \log \frac{P_i}{\pi_i} = H(P) + \mathrm{KL}(P \| \pi). \tag{4}$$

Since $H(P)$ is independent of $\theta$ and $\mathrm{KL}(P \| \pi) \geq 0$ with equality iff $\pi = P$, we have

$$\pi_\theta(\cdot \mid q, \boldsymbol{d}) \text{ minimizes } R(\boldsymbol{s}; q, \boldsymbol{d}) \iff \pi_\theta(\cdot \mid q, \boldsymbol{d}) = P(\cdot \mid q, \boldsymbol{d}). \tag{5}$$

Because $\pi_\theta(i \mid q, \boldsymbol{d}) = \frac{\exp(s(q,d_i))}{\sum_j \exp(s(q,d_j))}$ is a softmax over scores, any optimizer must satisfy

$$s(q, d_i) = \log P_i + C(q, \boldsymbol{d}), \tag{6}$$

for some additive constant $C(q, \boldsymbol{d})$ that is shared across all $i$ (hence irrelevant to the softmax).

**Step 3: Compute the posterior.** To compute $P_i$, note that by Bayes' rule and the sampling scheme,

$$P_i = \mathbb{P}(I = i \mid q, \boldsymbol{d}) \propto \mathbb{P}(I = i)\, p(q)\, p(d_i \mid I = i, q) \prod_{j \neq i} p(d_j \mid I = i, q) \tag{7}$$

$$= \frac{1}{K+1}\, p(q)\, p_{\mathrm{pos}}(d_i \mid q) \prod_{j \neq i} p(d_j), \tag{8}$$

where we used $p(d_j \mid I = i, q) = p(d_j)$ for $j \neq i$ and $p(d_i \mid I = i, q) = p_{\mathrm{pos}}(d_i \mid q)$. Normalizing over $i$ yields

$$\mathbb{P}(I = i \mid q, \boldsymbol{d}) = \frac{\dfrac{p_{\mathrm{pos}}(d_i \mid q)}{p(d_i)}}{\displaystyle\sum_{j=0}^{K} \dfrac{p_{\mathrm{pos}}(d_j \mid q)}{p(d_j)}}. \tag{9}$$

Taking logs and plugging into the optimality condition above, we obtain

$$s^*(q, d_i) = \log P_i + C(q, \boldsymbol{d}) \tag{10}$$

$$= \log \frac{p_{\mathrm{pos}}(d_i \mid q)}{p(d_i)} - \log\left(\sum_{j=0}^{K} \frac{p_{\mathrm{pos}}(d_j \mid q)}{p(d_j)}\right) + C(q, \boldsymbol{d}) \tag{11}$$

$$= \log \frac{p_{\mathrm{pos}}(q, d_i)}{p(q)\, p(d_i)} + \log p(q) - \log p_{\mathrm{pos}}(q) - \log\left(\sum_{j=0}^{K} \frac{p_{\mathrm{pos}}(d_j \mid q)}{p(d_j)}\right) + C(q, \boldsymbol{d}) \tag{12}$$

The last four terms are independent of $d$ (they depend only on $q$ or the batch $\boldsymbol{d}$). Since the softmax is invariant to adding any constant shared across candidates, they can be absorbed into a single additive constant. Hence the Bayes-optimal scorer is equivalently

$$s^*(q, d) = \log \frac{p_{\mathrm{pos}}(q, d)}{p(q)\, p(d)} + C, \tag{13}$$

for some constant $C$ that does not depend on $d$. This completes the proof.

**Remark.** If negatives are drawn from a distribution $p_{\text{neg}}(d)$ other than $p(d)$, the same derivation yields $s^*(q, d) = \log \frac{p_{\text{pos}}(d|q)}{p_{\text{neg}}(d)} + C$. In all cases, $s^*$ is unique up to adding any function of $q$. $\qquad\square$

### E.2 PROOF OF PROPOSITION 3

*Proof.* The goal is to show that the density ratio naturally decomposes into a semantic term and an artifact term; if the artifact distribution differs between positives and negatives, the artifact contribution cannot vanish.

We use uppercase letters (e.g., $M_d$, $A_d$) to denote random vectors, and lowercase $m_d, a_d$) for their realizations. The argument proceeds by a change of variables. If $T$ is further assumed to be $C^1$ and bijective onto its image, then

$$p_{\text{pos}}(q, m_d, a_d) = p_{\text{pos}}(q, d)\,|\det J_T(d)|^{-1}, \tag{14}$$

$$p(m_d, a_d) = p(d)\,|\det J_T(d)|^{-1}. \tag{15}$$

Thus,

$$\frac{p_{\text{pos}}(q, d)}{p(q)p(d)} = \frac{p_{\text{pos}}(q, m_d, a_d)}{p(q)p(m_d, a_d)}. \tag{16}$$

Applying the chain rule twice gives

$$\log \frac{p_{\text{pos}}(q, m_d, a_d)}{p(q)\,p(m_d, a_d)} = \left[\log p_{\text{pos}}(q \mid m_d, a_d) - \log p(q)\right] + \left[\log p_{\text{pos}}(m_d, a_d) - \log p(m_d, a_d)\right]. \tag{17}$$

Decompose further as $\log p_{\text{pos}}(m_d, a_d) = \log p_{\text{pos}}(m_d) + \log p_{\text{pos}}(a_d \mid m_d)$ and $\log p(m_d, a_d) = \log p(m_d) + \log p(a_d \mid m_d)$, and add–subtract $\log p_{\text{pos}}(q \mid m_d)$ to isolate the $(q, m_d)$ contribution:

$$\log \frac{p_{\text{pos}}(q, m_d, a_d)}{p(q)\,p(m_d, a_d)} = \underbrace{\left[\log p_{\text{pos}}(q \mid m_d) - \log p(q)\right] + \left[\log p_{\text{pos}}(m_d) - \log p(m_d)\right]}_{\phi(q, m_d)}$$
$$+ \underbrace{\left[\log p_{\text{pos}}(q \mid m_d, a_d) - \log p_{\text{pos}}(q \mid m_d)\right] + \left[\log p_{\text{pos}}(a \mid m_d) - \log p(a \mid m_d)\right]}_{\psi(a_d \mid q, m_d)}. \tag{18}$$

If $p_{\text{pos}}(a_d \mid q, m_d) \neq p(a_d \mid m_d)$ on a set of positive measure, then the artifact term $\psi$ cannot vanish.

Since $s^*(q, d) = \phi(q, m_d) + \psi(a_d \mid q, m_d) + C$ is a deterministic function of $(q, m_d, a_d)$, we have

$$H(s^* \mid q, m_d, a_d) = 0. \tag{19}$$

Here $H(\cdot \mid \cdot)$ denotes conditional Shannon entropy. We will make use of the identity

$$I(X; Z \mid Y) = H(Z \mid Y) - H(Z \mid X, Y)$$

for conditional mutual information.

If $A \mid (q, m_d)$ is non-degenerate and $\psi(\cdot \mid q, m_d)$ is non-constant, then the induced distribution of $s^*$ given $(q, m_d)$ is non-degenerate, i.e.,

$$H(s^* \mid q, m_d) > 0. \tag{20}$$

Applying the above identity yields

$$I(A; s^* \mid q, m_d) = H(s^* \mid q, m_d) - H(s^* \mid q, m_d, a_d) > 0, \tag{21}$$

which establishes the claim. $\qquad\square$

### E.3 Proof of Proposition 4

*Proof.* Let $T : \mathcal{D} \to \mathcal{M} \times \mathcal{A}$ be a $C^1$ bijection onto its image with $T(d) = (M_d, A_d)$, and let the passage encoder $h_d : \mathcal{D} \to \mathbb{R}^m$ be $C^1$. Define $g(m, a) := h_d(T^{-1}(m, a))$ and fix a reference $a_0 \in \mathcal{A}$. Then for $(m, a)$ near $(m, a_0)$,

$$g(m, a) = g(m, a_0) + J_a(m, a_0)(a - a_0) + r(m, a), \quad \|r(m, a)\| = o(\|a - a_0\|),$$

where $J_a(m, a_0) = \left[\partial g(m, a)/\partial a\right]_{a=a_0}$. Writing

$$h_{\text{sem}}(m) := g(m, a_0), \qquad h_{\text{art}}(a; m) := J_a(m, a_0)(a - a_0),$$

we obtain the local additive form

$$h_d(d) = h_{\text{sem}}(M_d) + h_{\text{art}}(A_d; M_d) + r(M_d, A_d).$$

At this point, we make a simplifying assumption: the Jacobian $J_a(m, a_0)$ does not substantially depend on $m$, or any residual dependence can be absorbed into the remainder term. Under this idealization we may write $h_{\text{art}}(a; m) \approx h_{\text{art}}(a)$.

Consequently, for a dot-product retriever $s_\theta(q, d) = \langle h_q(q), h_d(d) \rangle$,

$$s_\theta(q, d) = \underbrace{\langle h_q(q), h_{\text{sem}}(M_d) \rangle}_{\text{semantic}} + \underbrace{\langle h_q(q), h_{\text{art}}(A_d) \rangle}_{\text{artifact (linear)}} + \varepsilon(q, M_d, A_d), \tag{22}$$

where $\varepsilon(q, M_d, A_d) := \langle h_q(q), r(M_d, A_d) \rangle$ satisfies $\varepsilon(q, M_d, A_d) = o(\|A_d - a_0\|)$ as $\|A_d - a_0\| \to 0$. In other words, the remainder vanishes to first order and can be neglected in the idealized decomposition. $\square$

**Remark.** The argument relies on a local first-order approximation and a simplifying assumption on the artifact Jacobian. These approximations are introduced only to obtain a clearer analytical decomposition of semantic and artifact contributions. In the main text, we empirically examine whether artifact features can be linearly decodable from $h_d(d)$, providing evidence in support of this idealized view.

## F Additional Linguistic Analyses

In this appendix, we provide supplementary analyses promised in Section 4.1. Specifically, we report (i) additional effect-size analyses for the comparisons in the main text, and (ii) results on the other 13 datasets beyond MS MARCO.

### F.1 Effect-Size Analyses

We quantify the magnitude of linguistic differences using standard effect-size measures (Hedges' $g$ for mean differences) and report associated significance levels. These statistics complement the significance tests in the main paper by showing not only whether differences are significant but also their practical magnitude. Table 11 summarizes results on MS MARCO for two contrasts: (i) positives vs. the unannotated pool, and (ii) LLM-generated vs. human-written passages.

Table 11: Effect sizes (Hedges' $g$) and significance for linguistic feature comparisons on MS MARCO. Positive values indicate higher scores for the first group. $p$-values smaller than numerical precision are reported as $p < 10^{-15}$.

| Comparison | PPL ($g$) | IDF ($g$) | $p$-value |
|---|---|---|---|
| Positives vs. Unannotated | $-0.214$ | $+0.047$ | $< 10^{-15}$ |
| LLM vs. Human | $-0.274$ | $+0.145$ | $< 10^{-15}$ |

We observe that both comparisons yield highly significant differences despite modest effect sizes. For perplexity (PPL), positives are more fluent than the unannotated pool ($g = -0.214$), and LLM passages are even more fluent than human passages ($g = -0.274$). For IDF, the effects are smaller ($g = 0.047$ and $0.145$ respectively) but consistently positive, indicating that both positives and LLM rewrites exhibit slightly greater lexical specificity. Taken together, these results show that supervision and source type both introduce systematic, statistically robust shifts in linguistic features, even if the magnitudes are moderate.

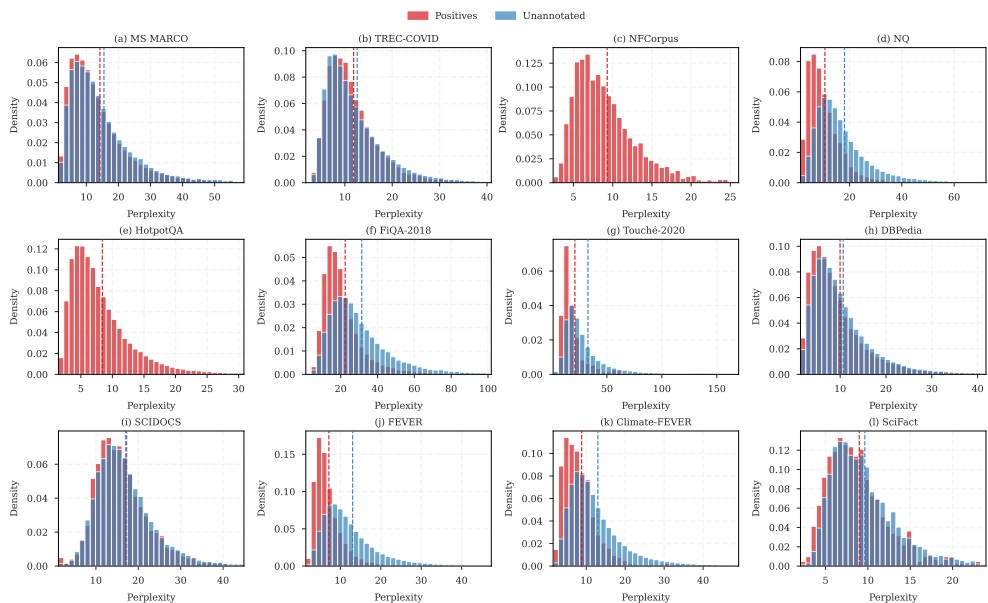

Figure 4: Perplexity distributions of positives versus negatives across retrieval datasets in Cocktail. For MS MARCO and DL19/20, results are reported once due to corpus overlap. For NFCorpus and HotpotQA, all passages are annotated as relevant, so only positive distributions are shown.

## F.2   POSITIVES VS. NEGATIVES ON ADDITIONAL DATASETS

To assess whether the imbalance between annotated positives and negatives generalizes beyond MS MARCO, we extend the perplexity analysis to other datasets in Cocktail (Figure4). For datasets that share the same corpus (e.g., MS MARCO and DL19/20), we report results only once. For NFCorpus and HotpotQA, all passages are annotated with relevance labels, so no negative pool exists and only positives are shown. Across the remaining datasets, positives consistently exhibit lower perplexity than negatives, mirroring the trend in MS MARCO. This indicates that stylistic disparities between positives and negatives are not dataset-specific idiosyncrasies but a systematic property of retrieval supervision. As discussed in the main text, positives are often drawn from edited, high-quality sources intended to serve as good answers, whereas negatives derive from more heterogeneous and less polished text.

## F.3   LLM VS. HUMAN ACROSS ADDITIONAL DATASETS

To ensure that the findings generalize beyond MS MARCO, we replicate the analysis on the other datasets in Cocktail. Figure 5 reports perplexity distributions, and Figure 6 reports IDF distributions, comparing LLM-generated versus human-written passages.

Consistent with the MS MARCO case, LLM-generated passages consistently exhibit lower perplexity and slightly higher IDF than their human-written counterparts. The PPL differences are stable and clear across all datasets, while the IDF differences are more modest in magnitude but follow the same direction throughout. These results confirm that source-based stylistic artifacts are systematic and broadly consistent across domains.

## G   COSINE SIMILARITY BETWEEN RANDOM HIGH-DIMENSIONAL VECTORS

We derive the null distribution of cosine similarities between independent random vectors, which serves as the statistical baseline for our embedding-space analyses. Let $x, y \in \mathbb{R}^m$ be isotropic random vectors. Normalizing to the unit sphere ($\hat{x} = x/\|x\|$, $\hat{y} = y/\|y\|$) yields $\hat{x}, \hat{y} \sim \mathrm{Unif}(\mathbb{S}^{m-1})$, and their cosine similarity is

$$Z = \langle \hat{x}, \hat{y} \rangle \in [-1, 1].$$

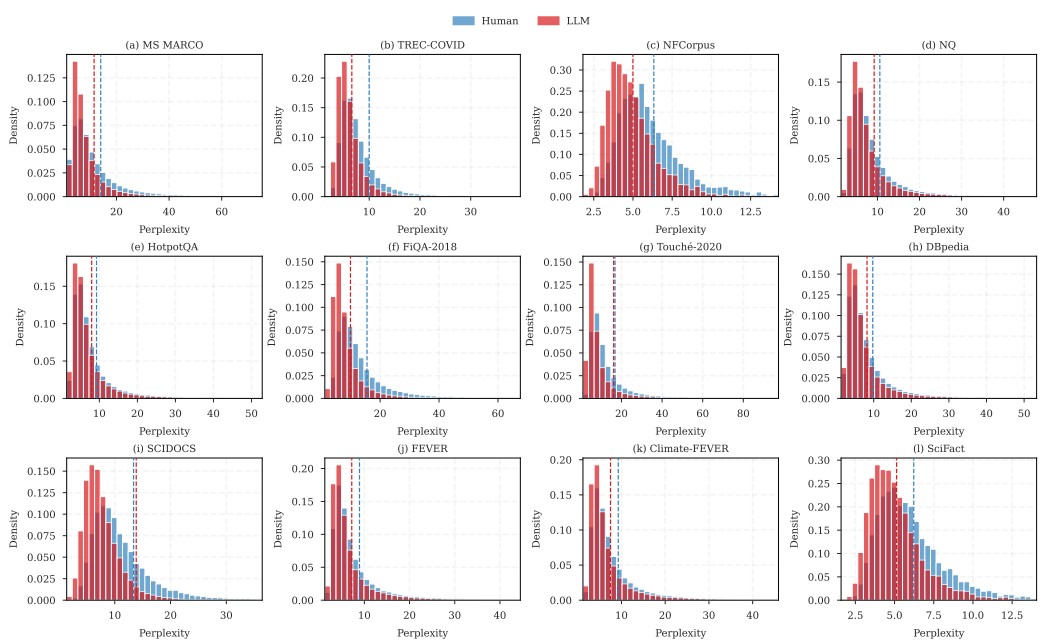

Figure 5: Perplexity (PPL) distributions of LLM-generated vs. human-written passages across additional datasets. Red = LLM, Blue = Human. LLM passages consistently exhibit lower perplexity, indicating higher fluency.

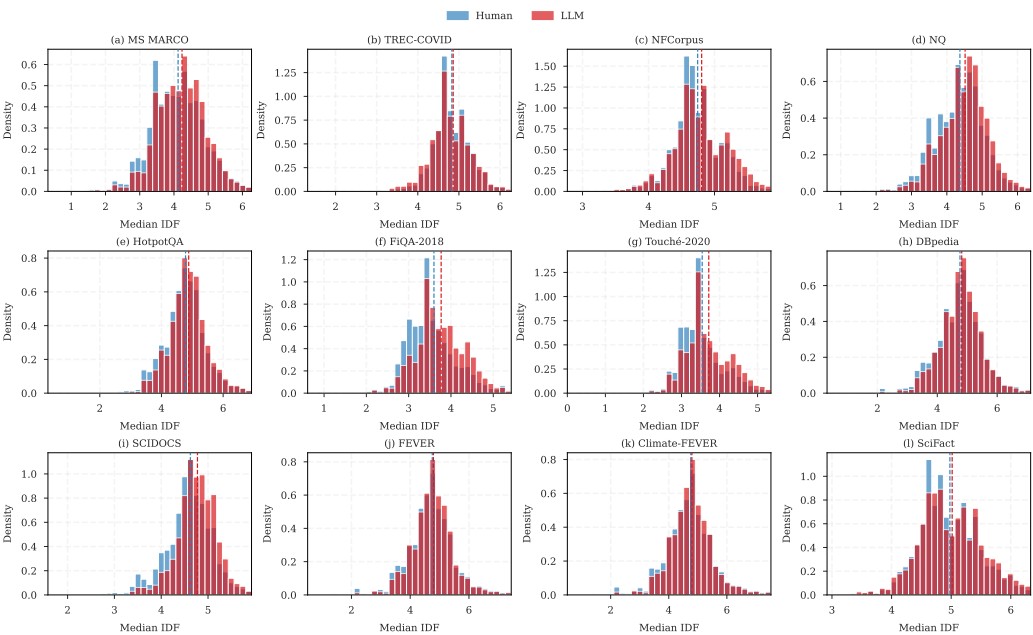

Figure 6: Median IDF distributions of LLM-generated vs. human-written passages across additional datasets. Red = LLM, Blue = Human. LLM passages generally exhibit higher IDF, though the gap varies across datasets.

By rotational invariance, $Z$ follows a Beta-type density (Vershynin, 2018):

$$f_Z(z) \;=\; \frac{\Gamma(\frac{m}{2})}{\sqrt{\pi}\,\Gamma(\frac{m-1}{2})}(1-z^2)^{\frac{m-3}{2}}, \quad z \in [-1, 1],$$

which is symmetric around zero. Equivalently, the tail probability can be expressed via the regularized incomplete Beta function:

$$\Pr(|Z| > t) \;=\; \mathrm{I}_{1-t^2}\!\left(\tfrac{m-1}{2}, \tfrac{1}{2}\right).$$

By symmetry, $\mathbb{E}[Z] = 0$. Since each coordinate of a uniform unit vector has variance $1/m$, the variance of $Z$ is

$$\mathrm{Var}(Z) \;=\; \frac{1}{m}.$$

For large $m$, the density concentrates sharply at zero. Expanding $\log(1 - z^2) \approx -z^2$ near the origin gives the Gaussian approximation

$$Z \;\approx\; \mathcal{N}\!\left(0, \tfrac{1}{m}\right).$$

In dimension $m = 768$, the standard deviation is $\sigma = 1/\sqrt{m} \approx 0.0361$, so that $3\sigma \approx 0.108$. Under the normal approximation,

$$\Pr(|Z| > 3\sigma) \approx 0.27\%,$$

which closely matches the exact Beta distribution. Thus, over 99.7% of random pairs fall within $\pm 3\sigma$, validating the use of this threshold as a significance criterion in high-dimensional embedding spaces. Figure 7 illustrates this concentration.

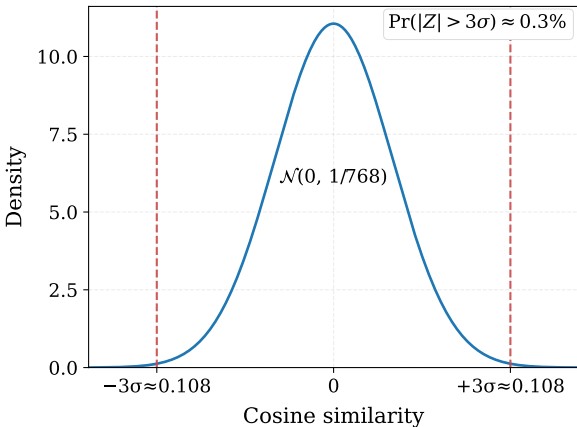

Figure 7: Null distribution of cosine similarity between random vectors in $m = 768$ dimensions, approximated by $\mathcal{N}(0, 1/m)$. Over 99.7% of values lie within $\pm 3\sigma \approx 0.108$, supporting its use as a significance criterion.

## H  ADDITIONAL EMBEDDING ANALYSES

In this appendix, we provide the full embedding-space analyses across all 12 distinct corpora in the Cocktail benchmark, using the DRAGON retriever as a representative model. Our experiments use 14 datasets from the Cocktail benchmark. Since three of them (MS MARCO, DL19, and DL20) share the same underlying corpus, we report embedding statistics at the corpus level, resulting in 12 unique corpora. These figures complement the representative results shown in the main text and report: (1) within-dataset displacement consistency (Figure 8), (2) cross-dataset similarity of mean displacement directions (Figure 9), and (3) alignment between LLM–human and supervision-induced directions (Figure 10).

Overall, these results extend the main-text findings to the full set of datasets. The majority of datasets follow the same trends as reported in the main text, while a small number exhibit weaker effects, which we discuss as exceptions rather than contradictions.

## I  ADDITIONAL RESULTS FOR RQ3

In this section, we provide the supplementary results for Section 5, including (a) retrieval effectiveness for the training-time sampling experiments, which were omitted from the main text due to space constraints, and (b) additional inference-time debiasing results on more datasets. These results complement the main findings and further validate our conclusions.

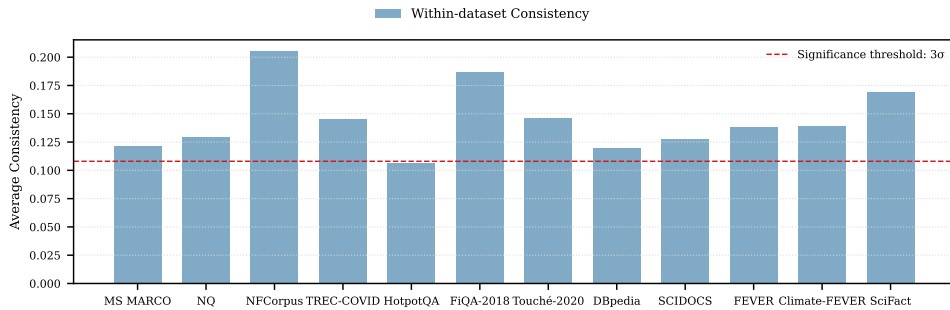

Figure 8: Within-dataset consistency of LLM–Human displacements. Bars show average pairwise cosine similarity among displacement vectors $\delta_i^{\text{LH}}$ within each dataset, relative to the $3\sigma$ significance threshold. Most datasets exceed the threshold, with a few exceptions near or below it.

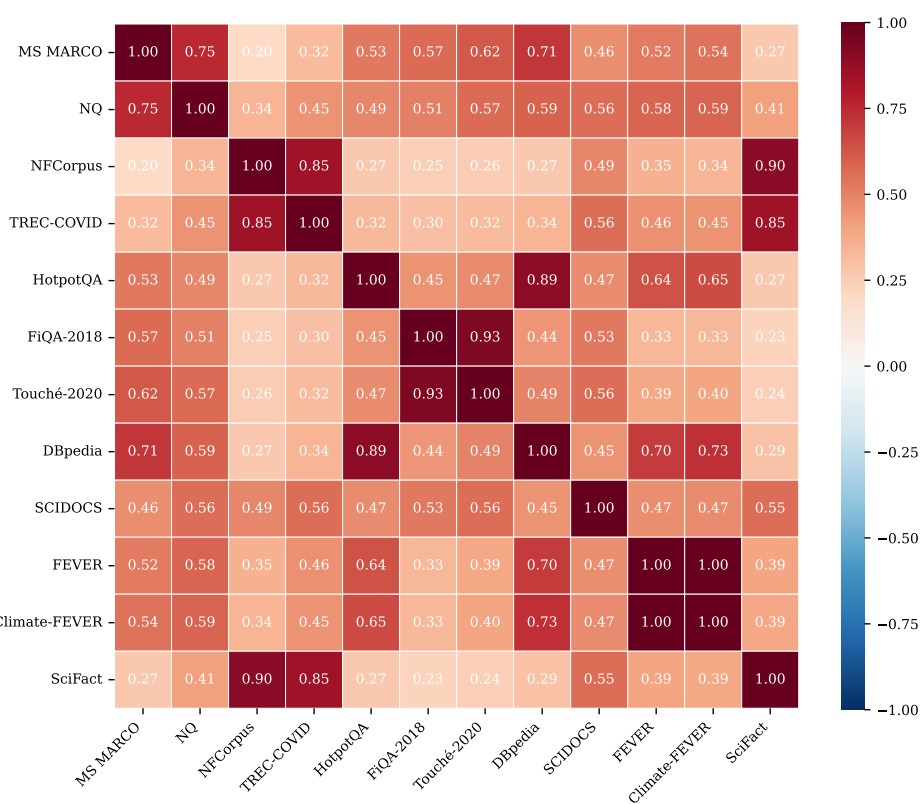

Figure 9: Cross-dataset similarity of mean LLM–Human displacement directions. Values denote cosine similarity between dataset-level means $\bar{\delta}_{\text{LH,D}}$. Darker cells indicate stronger alignment, revealing consistent artifact-induced directions across corpora.

## I.1 TRAINING-TIME INTERVENTIONS: RETRIEVAL EFFECTIVENESS

Table 12 reports retrieval effectiveness (NDCG@5) for the three negative sampling strategies (*in-batch only*, *standard*, and *hard-neg only*) across all datasets. Overall, settings that include mined hard negatives achieve higher retrieval performance, while using only in-batch negatives leads to lower effectiveness on most datasets. This trend is consistent with widely noted observations in the dense retrieval community that mined hard negatives are essential for strong retrieval effectiveness.

## I.2 INFERENCE-TIME INTERVENTIONS: ADDITIONAL DATASETS

We extend the inference-time evaluation beyond the five datasets shown in the main text. Table 13 reports $\Delta$NDSR@5 across all datasets, while Table 14 shows the corresponding NDCG@5 results.

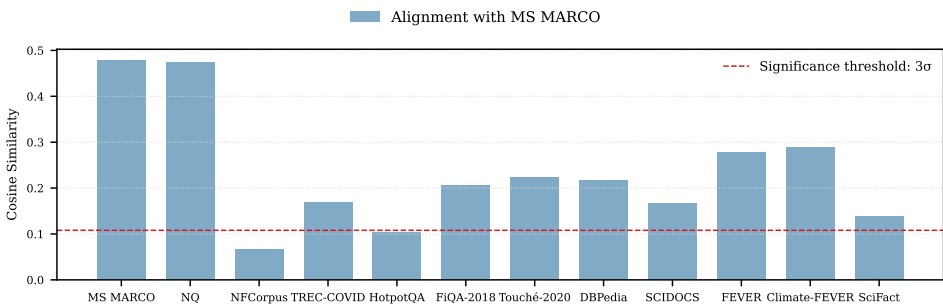

Figure 10: Cosine similarity between the LLM–Human displacement direction and the MS MARCO positive–negative contrast, across datasets. The red dashed line marks the $3\sigma$ significance threshold derived under the random null. Most datasets show strong alignment beyond the threshold, with a few cases near or below it.

Table 12: NDCG@5 results on 14 datasets under different negative sampling strategies. The "Standard" strategy combines in-batch and hard negatives, while the other two use only one type.

| Dataset | In-batch only | Standard | Hard-neg only |
|---|---|---|---|
| MS MARCO | 0.629 | 0.629 | 0.623 |
| DL19 | 0.640 | 0.706 | 0.728 |
| DL20 | 0.642 | 0.701 | 0.719 |
| TREC-COVID | 0.571 | 0.611 | 0.568 |
| NFCorpus | 0.303 | 0.287 | 0.278 |
| NQ | 0.652 | 0.670 | 0.666 |
| HotpotQA | 0.570 | 0.579 | 0.579 |
| FiQA-2018 | 0.209 | 0.218 | 0.216 |
| Touché-2020 | 0.350 | 0.418 | 0.411 |
| DBpedia | 0.428 | 0.436 | 0.437 |
| SCIDOCS | 0.096 | 0.086 | 0.086 |
| FEVER | 0.850 | 0.842 | 0.829 |
| Climate-FEVER | 0.280 | 0.271 | 0.241 |
| SciFact | 0.435 | 0.452 | 0.443 |
| **Average** | 0.475 | 0.493 | 0.487 |

Overall, the projection method generally reduces source bias, while retrieval effectiveness is largely preserved across datasets, consistent with the main text findings.

## J REVISITING LENGTH EFFECTS IN EVALUATING SOURCE BIAS

### J.1 LENGTH BIAS IN NEURAL RETRIEVAL

Prior work has shown that neural retrievers exhibit non-semantic bias correlated with passage length. On BEIR's Touché-2020 argument retrieval task, Thakur et al. (2024) reports that neural retrievers tend to rank very short passages, many of which are non-argumentative, leading to large drops in effectiveness. Complementary evidence documents neural retrievers exhibit systematic brevity, early-position, and literal-match biases, and that these biases often result in shorter passages receiving disproportionately high scores, even outranking passages that contain the correct answer (Fayyaz et al., 2025).

This consideration is particularly relevant for evaluating *source bias*, since differences between human-written and LLM-generated passages may be partially reflected in their length. The Cocktail benchmark (Dai et al., 2024a), widely used to compare LLM-generated and human-written passages, constructs LLM rewrites that are typically shorter. This leads to the central question we examine: *is the observed source bias simply a manifestation of length bias?*

Table 13: ΔNDSR@5 results (original vs. debiased) across 14 datasets and 5 relevance-supervised retrievers. Positive values indicate a preference for human-written passages, whereas negative values indicate a preference for LLM-generated ones. In the Average row, the first line reports the mean ΔNDSR@5, and the second line shows the remaining proportion of |ΔNDSR@5| after debiasing (original = 100%). Shading in the Average row reflects the relative magnitude of |ΔNDSR@5|, with darker colors indicating stronger source bias.

| Dataset (↓) | ANCE | | coCondenser | | DRAGON | | RetroMAE | | TAS-B | | % |
|---|---|---|---|---|---|---|---|---|---|---|---|
| | Original | Debias | Original | Debias | Original | Debias | Original | Debias | Original | Debias | |
| MS MARCO | -0.042 | 0.168 | -0.020 | 0.094 | -0.083 | -0.065 | -0.083 | 0.011 | -0.121 | -0.062 | 0 |
| DL19 | -0.073 | 0.197 | -0.072 | 0.096 | -0.233 | -0.160 | -0.186 | 0.076 | -0.224 | -0.151 | 10 |
| DL20 | -0.034 | 0.270 | -0.079 | 0.011 | -0.121 | -0.103 | -0.088 | 0.015 | -0.072 | 0.007 | 20 |
| TREC-COVID | -0.162 | -0.178 | -0.340 | -0.281 | -0.134 | -0.154 | -0.194 | -0.098 | -0.328 | -0.248 | 30 |
| NFCorpus | -0.087 | -0.067 | -0.068 | -0.064 | -0.079 | -0.064 | -0.081 | -0.044 | -0.082 | -0.057 | 40 |
| NQ | -0.042 | -0.032 | -0.072 | -0.071 | -0.099 | -0.085 | -0.060 | -0.044 | -0.078 | -0.062 | 50 |
| HotpotQA | -0.020 | 0.014 | -0.014 | 0.029 | -0.018 | -0.031 | -0.019 | 0.045 | -0.018 | -0.024 | 60 |
| FiQA-2018 | -0.179 | -0.159 | -0.219 | -0.263 | -0.161 | -0.154 | -0.205 | -0.201 | -0.170 | -0.182 | 70 |
| Touché-2020 | -0.168 | -0.148 | -0.226 | -0.153 | -0.178 | -0.162 | -0.175 | -0.127 | -0.247 | -0.197 | 80 |
| DBpedia | -0.097 | 0.025 | -0.054 | -0.015 | -0.057 | -0.055 | -0.059 | 0.006 | -0.042 | -0.036 | 90 |
| SCIDOCS | -0.040 | 0.069 | -0.058 | -0.053 | -0.048 | -0.012 | -0.073 | 0.007 | -0.054 | 0.010 | 100 |
| FEVER | -0.200 | -0.061 | -0.037 | -0.041 | -0.043 | -0.031 | -0.010 | 0.031 | -0.029 | -0.029 | |
| Climate-FEVER | -0.314 | -0.225 | -0.153 | -0.066 | -0.091 | -0.066 | -0.105 | 0.023 | -0.083 | -0.064 | |
| SciFact | -0.025 | -0.020 | -0.049 | -0.033 | -0.041 | -0.042 | -0.048 | -0.043 | -0.058 | -0.063 | |
| **Average** | -0.106 | -0.011 | -0.104 | -0.036 | -0.099 | -0.084 | -0.099 | -0.044 | -0.115 | -0.083 | |
| | (100%) | (10%) | (100%) | (35%) | (100%) | (85%) | (100%) | (44%) | (100%) | (72%) | |

Table 14: NDCG@5 results (original vs. debias) on 14 datasets for 5 relevance-supervised retrievers.

| Dataset (↓) | ANCE | | coCondenser | | DRAGON | | RetroMAE | | TAS-B | |
|---|---|---|---|---|---|---|---|---|---|---|
| | Original | Debias | Original | Debias | Original | Debias | Original | Debias | Original | Debias |
| MS MARCO | 0.590 | 0.568 | 0.620 | 0.621 | 0.665 | 0.665 | 0.626 | 0.626 | 0.617 | 0.617 |
| DL19 | 0.695 | 0.706 | 0.750 | 0.747 | 0.767 | 0.769 | 0.739 | 0.743 | 0.743 | 0.743 |
| DL20 | 0.716 | 0.671 | 0.750 | 0.751 | 0.778 | 0.779 | 0.760 | 0.771 | 0.737 | 0.740 |
| TREC-COVID | 0.679 | 0.690 | 0.707 | 0.695 | 0.684 | 0.681 | 0.744 | 0.737 | 0.644 | 0.638 |
| NFCorpus | 0.301 | 0.304 | 0.382 | 0.381 | 0.397 | 0.396 | 0.373 | 0.376 | 0.375 | 0.381 |
| NQ | 0.628 | 0.626 | 0.687 | 0.687 | 0.737 | 0.737 | 0.704 | 0.704 | 0.689 | 0.689 |
| HotpotQA | 0.537 | 0.537 | 0.640 | 0.639 | 0.719 | 0.719 | 0.716 | 0.715 | 0.674 | 0.673 |
| FiQA-2018 | 0.255 | 0.255 | 0.244 | 0.244 | 0.323 | 0.322 | 0.278 | 0.277 | 0.257 | 0.261 |
| Touché-2020 | 0.487 | 0.475 | 0.326 | 0.333 | 0.501 | 0.513 | 0.444 | 0.450 | 0.429 | 0.415 |
| DBpedia | 0.435 | 0.436 | 0.525 | 0.522 | 0.540 | 0.540 | 0.526 | 0.524 | 0.518 | 0.518 |
| SCIDOCS | 0.114 | 0.113 | 0.124 | 0.125 | 0.148 | 0.146 | 0.136 | 0.136 | 0.138 | 0.133 |
| FEVER | 0.824 | 0.829 | 0.785 | 0.786 | 0.895 | 0.894 | 0.891 | 0.892 | 0.858 | 0.858 |
| Climate-FEVER | 0.240 | 0.245 | 0.237 | 0.240 | 0.290 | 0.291 | 0.279 | 0.283 | 0.286 | 0.287 |
| SciFact | 0.429 | 0.427 | 0.530 | 0.526 | 0.599 | 0.595 | 0.571 | 0.574 | 0.564 | 0.563 |
| **Average** | 0.495 | 0.492 | 0.522 | 0.521 | 0.575 | 0.575 | 0.556 | 0.558 | 0.537 | 0.537 |

## J.2 EXTENDED REWRITING FOR LENGTH CONTROL

To answer this question, we require an evaluation setting in which passage *source* can be varied while passage *length* is manipulated independently of meaning. However, because the LLM-generated versions in Cocktail are generally shorter than the original human-written passages, they do not allow us to determine whether any observed preference for LLM-generated text reflects its source or simply its reduced length.

To control for length effects, we construct an additional rewritten version of each dataset that preserves the semantic content of the original passage while producing a systematically longer variant. Following the semantic-preserving rewriting protocol of previous work (Dai et al., 2024a), we use an LLM to regenerate each passage based solely on its original content, using a prompt that enforces meaning preservation while generating a slightly longer paraphrase. The full prompt is provided in Appendix J.5.

This yields three aligned versions for every dataset: the human-written version, the Cocktail version, and our length-extended version. These parallel versions enable controlled comparison in settings where rewritten passages are either shorter than or longer than their human counterparts. We report the resulting length statistics for all datasets in the next subsection.

Table 15: Average passage lengths across three versions of each dataset, along with relative changes ($\Delta$) with respect to the original human-written passages. For datasets sharing the MS MARCO corpus (e.g., DL19 and DL20), length statistics are reported once.

| Dataset | Human (Orig.) | Cocktail | Ours | $\Delta$ Cocktail (%) | $\Delta$ Ours (%) |
|---|---|---|---|---|---|
| MS MARCO | 59.1 | 55.8 | 82.0 | $-5.6$ | $+38.7$ |
| TREC-COVID | 204.8 | 171.8 | 222.0 | $-16.1$ | $+8.4$ |
| NFCorpus | 227.6 | 184.8 | 249.3 | $-18.8$ | $+9.5$ |
| NQ | 89.6 | 82.2 | 110.8 | $-8.2$ | $+23.6$ |
| HotpotQA | 68.5 | 67.4 | 92.9 | $-1.6$ | $+35.6$ |
| FiQA-2018 | 135.4 | 109.3 | 147.6 | $-19.3$ | $+9.0$ |
| Touché-2020 | 167.2 | 135.5 | 172.7 | $-18.9$ | $+3.3$ |
| DBpedia | 54.0 | 54.7 | 78.5 | $+1.3$ | $+45.3$ |
| SCIDOCS | 174.0 | 166.0 | 182.7 | $-4.6$ | $+5.0$ |
| FEVER | 99.4 | 91.9 | 118.7 | $-7.5$ | $+19.4$ |
| Climate-FEVER | 87.2 | 82.0 | 107.9 | $-6.0$ | $+23.8$ |
| SciFact | 209.4 | 178.9 | 230.2 | $-14.5$ | $+9.9$ |

Table 16: $\Delta$NDSR@5 results across 14 datasets for 13 neural retrievers spanning three model families. Negative values are shaded in red to indicate a preference for LLM-generated passages, while positive values are shaded in blue to indicate a preference for human-written passages. Asterisks (*) denote statistically significant deviations from zero (two-sided t-test, $p < 0.05$).

| Dataset ($\downarrow$) | Relevance-Supervised Retrievers | | | | | General-Purpose Embedding Models | | | | | Unsupervised Retrievers | | |
|---|---|---|---|---|---|---|---|---|---|---|---|---|---|
| | ANCE | TAS-B | coCondenser | RetroMAE | DRAGON | BGE | BCE | GTE | E5 | M3E | Contriever | E5-Unsup | SimCSE |
| MS MARCO | -0.026* | -0.042* | 0.032* | -0.014* | -0.007 | 0.040* | 0.383* | -0.038* | 0.157* | 0.446* | 0.271* | -0.113* | 0.658* |
| DL19 | -0.092 | -0.039 | -0.065* | -0.056 | -0.016 | 0.079 | 0.561* | -0.072 | 0.256* | 0.554* | 0.320* | -0.079 | 0.741* |
| DL20 | -0.007 | -0.033 | -0.029 | -0.020 | -0.039 | 0.212* | 0.458* | -0.004 | 0.227* | 0.648* | 0.367* | 0.017 | 0.793* |
| NQ | -0.050* | -0.045* | -0.035* | -0.002 | -0.047* | -0.085* | 0.498* | 0.024* | 0.313* | 0.320* | 0.197* | 0.152* | 0.369* |
| NFCorpus | -0.071* | -0.031* | -0.025* | -0.055* | -0.018 | -0.031* | 0.090* | -0.015 | 0.356* | 0.049* | -0.004 | -0.165* | 0.423* |
| TREC-COVID | -0.028 | -0.115 | -0.056 | -0.233* | -0.079 | -0.061 | 0.358* | -0.012 | 0.151* | 0.390* | -0.113* | 0.039 | 0.569* |
| HotpotQA | 0.024* | 0.067* | 0.024* | 0.034* | -0.002 | 0.151* | 0.354* | 0.051* | 0.238* | 0.237* | -0.484* | -0.044* | 0.110* |
| FiQA-2018 | -0.195* | -0.095* | -0.065* | -0.154* | -0.103* | -0.143* | 0.607* | -0.016 | -0.085* | 0.414* | 0.085* | -0.063* | 0.513* |
| Touche-2020 | 0.016 | -0.010 | -0.050 | -0.002 | -0.022 | -0.055 | 0.667* | 0.022 | -0.168* | 0.636* | -0.101 | 0.003 | 0.268* |
| DBPedia | -0.033 | -0.017 | -0.018 | -0.012 | -0.023 | 0.112* | 0.369* | 0.046* | 0.242* | 0.444* | -0.314* | 0.004 | 0.371* |
| SCIDOCS | -0.023* | -0.075* | -0.074* | -0.007 | -0.024* | -0.148* | 0.572* | -0.019* | 0.188* | 0.453* | 0.159* | 0.032* | 0.386* |
| FEVER | -0.155* | -0.034* | -0.021* | -0.016* | -0.021* | 0.037* | 0.442* | 0.061* | 0.159* | 0.247* | -0.075* | 0.066* | 0.105* |
| Climate-FEVER | -0.245* | -0.103* | -0.064* | -0.147* | -0.132* | -0.127* | 0.742* | -0.022* | 0.487* | 0.253* | -0.027* | 0.147* | 0.068* |
| SciFact | -0.084* | -0.027 | -0.041* | -0.010 | -0.011 | -0.016 | -0.006 | -0.021 | 0.133* | -0.059* | 0.173* | -0.071* | 0.181* |

## J.3 LENGTH STATISTICS ACROSS THE THREE VERSIONS

Table 15 reports the average passage lengths across the three versions. Consistent with prior observations, the Cocktail version reduces passage length by approximately 5–20% across datasets. By contrast, our extended version increases length by 3–45%. Taken together, these variants provide controlled conditions in which LLM-generated passages can be either shorter or longer than their human-written counterparts.

The length statistics in our analysis are computed using the standard Apache Lucene tokenizer without stemming or stopword removal. As tokenization procedures differ across implementations, our computed averages show small deviations from the values reported in the Cocktail benchmark (Dai et al., 2024a). For completeness, we include those reported values in Table 8.

## J.4 DOES LENGTH ALONE ACCOUNT FOR RETRIEVER PREFERENCES?

As shown in Section 3, source bias is not a universal property of neural retrievers: it appears consistently only in models trained with relevance supervision, whereas unsupervised and general-purpose embedding models do not exhibit systematic source preference. A natural follow-up question is whether these supervised models seem to favor LLM-generated passages simply because the LLM passages in Cocktail are, on average, shorter than their human-written counterparts.

To assess whether length differences alone can account for the observed preferences, we repeat the source-preference evaluation on the length-controlled versions introduced above. In this setting, each LLM passage is a meaning-preserving yet systematically lengthened version of its human-written counterpart. This setup removes the original brevity advantage of LLM passages in Cocktail

Table 17: ΔNDSR@5 results of unsupervised retrievers after MS MARCO fine-tuning, corresponding to the same base models in Table 16. The "-FT" suffix denotes fine-tuning on MS MARCO. Negative values are shaded in red to indicate a preference for LLM-generated passages, while positive values are shaded in blue to indicate a preference for human-written passages. Asterisks (*) denote statistically significant deviations from zero (two-sided t-test, $p < 0.05$).

| Dataset (↓) | Relevance-Supervised Retrievers | | |
|---|---|---|---|
| | Contriever-FT | E5-FT | SimCSE-FT |
| MS MARCO | 0.110* | -0.008* | -0.020* |
| DL19 | -0.015 | -0.046 | -0.082 |
| DL20 | 0.218* | 0.071 | -0.021 |
| NQ | -0.017* | -0.049* | -0.048* |
| NFCorpus | -0.007 | -0.083* | -0.019 |
| TREC-COVID | -0.238* | -0.242* | -0.036 |
| HotpotQA | 0.111* | -0.012* | 0.035* |
| FiQA-2018 | -0.016 | -0.123* | -0.206* |
| Touche-2020 | -0.041 | -0.042 | -0.089 |
| DBPedia | 0.060 | -0.017 | -0.044* |
| SCIDOCS | -0.026* | -0.022* | -0.056* |
| FEVER | 0.078* | -0.041* | 0.029* |
| Climate-FEVER | -0.091* | -0.145* | -0.120* |
| SciFact | -0.021 | -0.058* | -0.052 |

and, in most cases, even places them at a length disadvantage. Under these conditions, any remaining preference for LLM-generated text cannot be attributed to shorter passage length.

Table 16 shows the results for the full set of 13 retrievers. Among the relevance-supervised models, lengthening the LLM passages reduces the magnitude of the bias, often substantially, but does not eliminate it. These models continue to prefer LLM-generated passages on most datasets, despite the fact that the LLM versions are now longer than the human counterparts. A few model–dataset pairs even flip to a slight preference for human-written passages, but the dominant pattern remains a persistent bias toward LLM passages. This indicates that passage length modulates the strength of the effect but is insufficient to account for its origin.

Table 17 further examines the three unsupervised retrievers from Section 3 after MS MARCO fine-tuning. Models that originally exhibited little or no source preference develop a consistent bias toward LLM-generated passages once trained with relevance supervision—even when the LLM passages face the same length disadvantage. In contrast, general-purpose embedding models and the unsupervised retrievers in their base form tend to favor the human passages under the length-controlled condition, reflecting their greater sensitivity to length but also their lack of a stable preference in either direction.

Taken together, these findings suggest that although passage length influences the strength of source preference, it is insufficient to explain why the effect persists. Lengthening the LLM passages attenuates the bias but does not remove it, and relevance supervision still induces a preference for LLM-generated text when length advantages are reversed. This aligns with the overall conclusion that source bias primarily arises from stylistic and distributional artifacts in supervised retrieval data.

### J.5 PROMPT FOR LENGTH-CONTROLLED REWRITING

To produce the length-controlled passages used in our evaluation, we employ a prompt that enforces meaning preservation while generating slightly longer variants of the original text. The full prompt template is shown in Figure 11. While the prompt specifies a modest increase (e.g., around 10%), LLM outputs exhibit natural variability, and some rewritten passages become substantially longer. This variability does not affect our analysis, as the experiment requires only that the LLM passages be consistently longer than the originals, not that they match a specific percentage.

## K ADDITIONAL ANALYSES OF SUPERVISION AND NEGATIVE SAMPLING

This section provides further evidence on how different *supervision pipelines* and *negative sampling strategies* influence source bias. Our goal is not to analyze negative sampling in isolation, but to understand how supervision design induces (or avoids) *artifact imbalances* between positives and negatives, a central component of the mechanism developed in Section 4.

```
<|system|>
You are a helpful assistant.

<|user|>
Please follow the instructions below:
1. Maintain the original meaning of the input passage.
2. Make the paraphrased passage slightly longer than the original (e.g., 10% longer) while
     preserving the same information.
3. Output the paraphrased passage directly.

Following is the passage you need to paraphrase:
{text}
Your answer must be formatted as:
Rewritten Text:
<your rewritten text>
```

Figure 11: Prompt used to generate meaning-preserving, length-extended passages with Qwen2.5-7B-Instruct.

This section begins by clarifying how different supervised retrievers construct their hard negatives, since these pipelines determine the stylistic differences between positive and negative passages. We then evaluate a retriever trained solely on NQ to examine whether the observed source bias generalizes beyond MS MARCO. Finally, we evaluate a more practical alternative to the in-batch-only setup by selecting hard negatives directly from the positive pool.

### K.1 SUPERVISION PIPELINES AND HARD-NEGATIVE CONSTRUCTION

Relevance-supervised retrievers (e.g., ANCE, TAS-B, DRAGON, RetroMAE, coCondenser) typically rely on large-scale human relevance annotations, coupled with mined hard negatives. Although their exact pipelines differ, they all follow a similar supervised contrastive setup: for each query $q$, a judged relevant passage $p^+$ is paired with one or more hard negatives $p^-$ drawn from top-ranked retrieval candidates that are either unjudged or explicitly labeled non-relevant.

Concretely, ANCE, coCondenser and DRAGON iteratively refresh hard negatives using the current dense retriever, while TAS-B and RetroMAE mainly rely on BM25- or multi-retriever–mined candidates from MS MARCO. In all cases, positives and negatives are sampled from *different parts of the corpus*: positives concentrate on answer-like spans around annotated answers, whereas negatives come from a much broader and noisier pool.

A common pattern therefore emerges:

- **Positives** are drawn from *answer-like, high-quality passages*, often with higher fluency and information density.
- **Negatives** are drawn from *retrieval candidates* (e.g., BM25 top-$k$ or model-mined candidates) that are more heterogeneous, noisier, and stylistically less polished.

As shown in Figure 1a and Appendix F, this supervision scheme induces a systematic stylistic gap between positive and negative pools. If we let $A$ denote a stylistic attribute (e.g., fluency), and write

$$\Delta A \;=\; \mathbb{E}[A^+] - \mathbb{E}[A^-],$$

then in typical relevance-supervised training we observe $\Delta A > 0$ across multiple artifact dimensions. This helps explain why *all relevance-supervised retrievers* in Table 1 exhibit pronounced source bias: supervision itself encodes a non-semantic separation between positives and negatives that overlaps with properties of LLM-generated text.

In contrast, the other model families in Table 1 use supervision schemes that do *not* introduce such a consistent gap:

- **General-purpose embedding models** (e.g., E5, BGE, GTE, M3E) are trained with multi-task and weakly supervised objectives, often with symmetric sampling from the same underlying corpus. Positives and negatives are drawn from similar distributions (e.g., STS/NLI pairs, contrastive sentence pairs), which keeps $\Delta A$ small.
- **Unsupervised retrievers** (e.g., Contriever, unsupervised SimCSE) sample positives via augmentation (e.g., dropout views, adjacent sentences) and use in-batch samples as negatives. Both come from the same raw corpus, again implying $\Delta A \approx 0$.

Table 18: Source bias of the NQ-only DPR retriever on Cocktail. We report $\Delta$NDSR@5; negative values indicate a preference for LLM-generated passages and are shaded in red. Asterisks (*) denote statistically significant deviations from zero (two-sided t-test, $p < 0.05$).

| Dataset | $\Delta$NDSR@5 | Dataset | $\Delta$NDSR@5 |
|---|---|---|---|
| MS MARCO | -0.264* | FiQA-2018 | -0.631* |
| DL19 | -0.258* | Touche-2020 | -0.430* |
| DL20 | -0.319* | DBPedia | -0.087* |
| NQ | -0.106* | SCIDOCS | -0.353* |
| NFCorpus | -0.170* | FEVER | -0.075* |
| TREC-COVID | -0.503* | Climate-FEVER | -0.119* |
| HotpotQA | -0.065* | SciFact | -0.300* |

Table 19: Comparison of negative sampling strategies. We report average $\Delta$NDSR@5 (bias; lower magnitudes indicate weaker source bias) and NDCG@5 (retrieval effectiveness) across the 14 Cocktail datasets. The "Positive-pool hard negs" condition retains hard negatives while reducing the stylistic gap between positive and negative pools.

| Training configuration | $\Delta$NDSR@5 | NDCG@5 |
|---|---|---|
| Standard | -0.099 | 0.493 |
| In-batch only | -0.024 | 0.475 |
| Positive-pool hard negs | -0.023 | 0.478 |

These differences can therefore be unified under a single mechanism: *whether the supervised training process creates a systematic stylistic imbalance between positives and negatives.*

### K.2 SOURCE BIAS PERSISTS UNDER NQ-ONLY SUPERVISION

A natural question is whether source bias is specific to MS MARCO and its particular negative-construction scheme. To test this, we evaluate a publicly available DPR retriever trained solely on Natural Questions (NQ)[1]. NQ differs from MS MARCO in several fundamental respects, including its Wikipedia-only domain, its distinct annotation and answer format, the retrieval candidates used for negative sampling, and the overall style and distribution of passages.

We evaluate this NQ-only retriever on Cocktail. Despite the differences in corpus, annotation, and negative sampling pipeline, it still exhibits substantial source bias. On average across the 14 datasets we observe $\Delta$NDSR@5 $= -0.263$, and the per-dataset results are shown in Table 18. All datasets show a clear preference for LLM-generated passages.

Although the magnitude varies across datasets, the direction of the effect mirrors that of MS MARCO-trained retrievers. This indicates that source bias is not an artifact of MS MARCO, but a more general outcome of relevance supervision. Whenever a supervised dataset introduces a stylistic discrepancy between positive and negative passages (i.e., $\Delta A > 0$), the resulting retriever tends to inherit a corresponding preference that favors LLM-generated text.

### K.3 FROM IN-BATCH NEGATIVES TO HARD NEGATIVES FROM THE POSITIVE POOL

Section 5 showed that training-time negative sampling has a direct and monotonic effect on source bias: moving from in-batch only to standard sampling to hard-neg only progressively increases artifact imbalance $\Delta A$ and strengthens bias (Table 3). The in-batch-only configuration serves as a useful *mechanism probe* because it removes all mined hard negatives and thus approximates the idealized condition $\Delta A \approx 0$. However, this setup is stricter than what typical retrieval systems would use in practice. To explore a more practical alternative, we examine a variant that *retains hard negatives* while reducing the stylistic gap between positive and negative pools.

To bridge this gap, we evaluate a more realistic variant that keeps hard negatives while reducing the stylistic gap between positives and negatives. For each annotated positive passage, we run BM25

---

[1]We use the publicly released DPR encoders trained on NQ: the question encoder (https://huggingface.co/sentence-transformers/facebook-dpr-question_encoder-single-nq-base) and the context encoder (https://huggingface.co/sentence-transformers/facebook-dpr-ctx_encoder-single-nq-base).

Table 20: Per-dataset $\Delta$NDSR@5 and NDCG@5 results for the "positive-pool hard negs" configuration.

| Dataset | $\Delta$NDSR@5 | NDCG@5 |
|---|---|---|
| MS MARCO | 0.027 | 0.621 |
| DL19 | -0.013 | 0.667 |
| DL20 | 0.012 | 0.661 |
| NQ | 0.019 | 0.668 |
| NFCorpus | -0.060 | 0.239 |
| TREC-COVID | -0.158 | 0.574 |
| HotpotQA | 0.013 | 0.577 |
| FiQA-2018 | 0.005 | 0.215 |
| Touche-2020 | -0.064 | 0.396 |
| DBPedia | -0.033 | 0.430 |
| SCIDOCS | -0.017 | 0.091 |
| FEVER | 0.037 | 0.844 |
| Climate-FEVER | -0.069 | 0.268 |
| SciFact | -0.023 | 0.436 |
| Average | -0.023 | 0.478 |

over the positive pool and select the highest-ranked non-ground-truth passages as hard negatives. These negatives are still semantically challenging but remain stylistically similar to positives, since both are drawn from the same answer-like distribution. Intuitively, this construction preserves hard negative difficulty while shrinking $\Delta A$.

Table 19 compares three settings for a representative relevance-supervised retriever: (1) the "Standard" configuration (in-batch + mined hard negatives), (2) the "In-batch only" setting from Section 5, and (3) the "Positive-pool hard negs" configuration (in-batch + BM25 negatives over positives). We report average $\Delta$NDSR@5 and NDCG@5 across all 14 datasets, and the full per-dataset metrics for the positive-pool variant are provided in Table 20.

Empirically, "In-batch" only yields the strongest reduction in source bias but also a drop in NDCG@5, consistent with prior observations that mined hard negatives help retrieval effectiveness. The "Positive-pool hard negs" setting achieves a similar suppression of bias as the in-batch-only condition, yet recovers much of the NDCG@5 lost in that extreme setup.

These results sharpen the role of negative sampling. First, *hard negatives themselves are not the cause of source bias*: models trained with positive-pool hard negatives still see difficult negatives but remain weakly biased. Instead, bias tracks the stylistic alignment between positive and negative pools: when both are drawn from similar stylistic distributions (small $\Delta A$), source bias is substantially reduced, even in the presence of hard negatives. Together with the NQ-only supervision results in Appendix K.2, this provides additional evidence that artifact imbalance in supervision, rather than the use of hard negatives by itself, is the primary driver of source bias.

