# OpenReview forum: "Data, Not Model: Explaining Bias toward LLM Texts in Neural Retrievers"
_ICLR.cc/2026/Conference — Submitted to ICLR 2026_

### Official Review · Reviewer_HiGy · 2025-10-28

**Soundness:** 3
**Presentation:** 3
**Contribution:** 3
**Rating:** 6
**Confidence:** 4

**Summary:**

This paper investigates the phenomenon of source bias, where neural retrievers preferentially rank text generated by LLMs over semantically similar human-written text. The authors challenge the prevailing view that this bias is an inherent flaw of retriever models. Instead, they posit that the bias originates from systematic, non-semantic artifacts within the supervised training data. They show that this bias is prominent only in models trained with relevance supervision, as the stylistic properties of positive documents mirror those of LLM-generated text, creating a learnable shortcut. This mechanism is supported by linguistic analysis, embedding-space geometry, and a formal theoretical framework. The paper validates this hypothesis by proposing two successful mitigation strategies: controlling artifact imbalance during training via negative sampling and removing the bias direction from embeddings at inference time.

**Strengths:**

* **Strong contribution**: The central argument that source bias is learned from data artifacts is a compelling and important shift from previous explanations focused on model properties.
* **Comprehensive evidentiary support**: The authors defend their thesis with a powerful combination of controlled experiments.
* **Good structure and clarity**: The organization around three distinct research questions makes the paper clear and the narrative easy to follow.
* **Practical mitigation strategies**: The proposed training- and inference-time techniques are not only practical solutions but also serve as an additional validation of the paper's core hypothesis.

**Weaknesses:**

* **Unsupported assertions**: The paper makes several strong claims without sufficient evidence or citations. For instance, it asserts that Llama-3-8B provides a "reliable proxy for human-perceived fluency" and that the presence of false negatives in retrieval datasets is "mostly irrelevant in practice", yet no supporting references are provided.
Impact of the relevance assessment method: Annotations highly dependent on how the candidate passages were retrieved (BM25 for MS MARCO vs. encoder-based for more recent datasets), more details should be given on that.
* **Lack of systematic statistical significance assessment**: Some values in Table 1 are close to 0, but no statistical significance testing is reported.
* **Limited analysis of linguistic features**: In Figure 1, while perplexity and IDF are informative, the examination of stylistic artifacts could be strengthened by incorporating additional metrics, such as passage length (in tokens or characters) or lexical diversity.
* **Absence of non-English data**: The study would benefit from analyzing at least one language other than English to explore cross-linguistic lexical features.
* **Typo**: Figure 3 is referenced in lines 319–320, but it does not appear in the paper (possibly referring to Figure 4).

**Questions:**

* **Unsupported assertions**: Can the authors provide empirical evidence or references to support the claims that Llama-3-8B serves as a "reliable proxy for human-perceived fluency" and that false negatives in retrieval datasets are "mostly irrelevant in practice"?
* **Impact of relevance assessment method**: Annotating retrieval datasets typically involves using a pre-retrieval step to collect candidate documents. How does the choice of this retrieval method (e.g., BM25 vs. encoder-based retrieval) influence the resulting relevance annotations and reported performance? It would be helpful to provide more details on this dependency.
* **Statistical significance assessment**: Were any statistical significance tests conducted for the values reported in Table 1, especially those close to 0? If not, can the authors clarify whether these differences are meaningful?
* **Linguistic features**: Could the analysis of stylistic artifacts be extended to include additional linguistic metrics, such as passage length (in tokens or characters) or lexical diversity, to provide a more comprehensive view?
* **Non-English data**: Would the authors consider including non-English data to assess whether the reported findings generalize across languages?

---

> ### Author Response · Authors · 2025-11-28
> **Author Response (1/2)**
>
> We thank the reviewer for the careful assessment and constructive feedback. Below, we respond to each comment (W1–W5, Q1–Q5) in turn and will incorporate the suggested clarifications and corrections into the revised manuscript.
>
> **1. Response to W1 + Q1: On the claims regarding fluency proxies and false negatives**
>
> **(1). Llama-3-8B as a fluency proxy.** We agree that the original phrasing could be interpreted more strongly than intended. Our intended meaning was only that LLM judges provide stable, reproducible fluency signals, not that they serve as a substitute for human evaluation. This practice is widely adopted in recent evaluation pipelines (e.g., MT-Bench and AlpacaEval), where LLM-based judges have been shown to produce stable fluency assessments and correlate well with human preference trends ([1], [2]). We have revised the wording to “useful automatic fluency measure” and added the appropriate references in the updated version.
>
> **(2) On the impact of false negatives.** We apologize for the confusion caused by the ambiguous pronoun reference in our original phrasing.
>
> > **Original Context:** "...Although the negative pool may contain unannotated false negatives, **it** is mostly irrelevant in practice."
>
> **Clarification:** The pronoun "**it**" was intended to refer to the **statistical impact** of false negatives on our specific linguistic analysis, rather than implying that false negatives themselves are irrelevant to retrieval performance. We fully agree that false negatives are critical in retrieval evaluation. However, in the context of analyzing linguistic features, given the massive size of the negative pool (e.g., 8.8M in MS MARCO) versus the sparse relevance labels, the probability of a randomly selected unlabelled document being relevant is negligible ([3]). Therefore, the presence of sparse false negatives does not statistically alter the aggregate linguistic distribution (e.g., PPL, IDF) of the non-relevant class.
>
> We have rewritten this sentence to be precise: "While the negative pool contains sparse false negatives, the majority are non-relevant, making it a representative sample for linguistic analysis."
>
> [1] Judging LLM-as-a-Judge: MT-Bench for Evaluating Chat Assistants. Zheng et al. NeurIPS 2023.
> [2] Length-Controlled AlpacaEval: A Simple Way to Debias Automatic Evaluators. Dubois et al. COLM 2024.
> [3] MS MARCO: A Human Generated MAchine Reading COmprehension Dataset.Nguyen, Tri, et al.
>
> **2. Response to W1 + Q2: Impact of relevance assessment method**
>
> We appreciate the reviewer for raising this insightful question. Large-scale IR datasets typically use a pre-retrieval stage (such as BM25 or production rankers) to form candidate pools, and this step influences which documents are presented to annotators. While pre-retrieval determines the candidates, our analysis examines the distribution that results from annotation. Specifically, we compare passages marked as relevant with the remaining passages in the corpus.
>
> In our view, the stylistic patterns in positives are more likely explained by annotation tendencies than by a particular pre-retrieval method, although both may play a role. Similar positive–negative stylistic differences appear across 14 BEIR-derived datasets with diverse construction pipelines, suggesting that such imbalance reflects general properties of relevance annotation rather than an artifact of one retrieval approach. We see this as an interesting nuance in how different stages of dataset construction shape IR benchmarks.
>
> **3. Response to W2 + Q3: Statistical significance assessment**
>
> We appreciate the reviewer’s suggestion. We have now added statistical significance tests to Table 1 using standard two-sided t-tests (p < 0.05), and the revised version has been uploaded.

---

> ### Author Response · Authors · 2025-11-28
> **Author Response (2/2)**
>
> **4. Response to W3 + Q4: Linguistic features**
>
> We thank the reviewer for the suggestion to examine additional linguistic features (e.g., length or lexical diversity). Following this direction, we extended our analysis using Formality ([1]), a widely used POS-based metric that captures the structured, objective, and “textbook-like” phrasing characteristic of the surface-level artifacts implicated in source bias. This makes Formality a suitable choice for quantifying such stylistic tendencies.
>
> | Dataset      | Comparison Pair         | Formality Score|
> | ------------ | ----------------------- | ------------------------- |
> | **MS MARCO** | Human Pos vs. Human Neg | **64.62** vs. 63.18       |
> | **MS MARCO** | LLM-Gen vs. Human       | **63.70** vs. 63.30       |
> | **NQ**       | LLM-Gen vs. Human       | **61.61** vs. 60.78       |
> | **FiQA**     | LLM-Gen vs. Human       | **52.60** vs. 47.69       |
> | **SciFact**  | LLM-Gen vs. Human       | **65.65** vs. 65.55       |
>
> Across datasets, positive passages consistently exhibit slightly higher formality, matching the trend seen in perplexity and IDF. The pattern persists across domains, supporting our claim that supervised relevance data encode subtle but consistent stylistic biases.
>
> [1]  Formality of language: definition, measurement and behavioral determinants[J]. Heylighen F, Dewaele J M. Interner Bericht, Center “Leo Apostel”, Vrije Universiteit Brüssel, 1999, 4(1).
>
> **5. Response to W4 + Q5: Non-English data**
>
> We appreciate this suggestion. While our current experiments focus on English, we believe the underlying mechanism is unlikely to be language-specific. The source bias we study arises from two factors that are broadly observed across languages: First, human annotators in many multilingual IR settings tend to prefer passages that are more fluent, well-structured, or answer-like when assigning relevance labels. Second, multilingual LLMs exhibit similar stylistic tendencies across languages, often producing text with high local coherence and low perplexity.
>
> Given these shared tendencies, the alignment between positive passages and LLM-generated styles reflects a general property of supervised relevance training rather than a phenomenon tied to English grammar.
>
> **6. Response to W5: Typo**
>
> Thank you for catching the figure reference issue. This has been corrected in the updated submission (the previous “Figure 4a/4b” is now correctly labeled “Figure 3a/3b”).

---

### Official Review · Reviewer_LRJy · 2025-10-30

**Soundness:** 2
**Presentation:** 3
**Contribution:** 2
**Rating:** 2
**Confidence:** 4

**Summary:**

The paper evaluates the setup of whether neural retrievers prefer LLM-text over human, presenting source bias that stems from supervised training datasets (MS MARCO). They evaluate dense retrieval models on the Cocktail benchmark and further suggest two techniques (training-time and inference-time) to reduce the source bias.

**Strengths:**

- The paper works on an interesting issue of the source bias towards LLM-generated texts over human texts, and ties the problem with training datasets instead of the model. The problem (source bias in retrieval models) has been studied previously.
- The paper evaluates multiple dense retrieval models, spanning across various families.
- The paper is well-written and overall easy to follow through.

**Weaknesses:**

Here I've listed below the weaknesses associated with the paper:

### **W1: Limited novelty in the work**

I find limited novelty in this work; the experiments have been conducted with existing dense retrieval models, evaluating them on an already published Cocktail benchmark (Dai et al. 2024), plus further studies in (Dai et al. 2024c) already reveal a source bias of neural retrieval models. This paper only extends the artifact seen towards "data" and not models, as the title suggests.

### **W2: The evaluation setup on Cocktail may not accurately reflect source bias towards LLM-generated texts.**

The evaluation results are solely based on the Cocktail benchmark and are not convincing. Here are three reasons why:

The cocktail benchmark (Dai et al. 2024) rewrites human-written passages from the BEIR benchmark (Thakur et al. 2021) using the Llama-2-7B model. The LLM-rewritten model is very similar to the original passage. However, there are two things which rewriting passage does: (a) removes grammatical mistakes, (b) removes unused words and bad punctuation present in text (e.g., [1], [2], present in Natural Questions), and most importantly (c) reduces the human passage content length on average (confirmed the lengths from Table 8).

First, I suspect the preference comes from (c) and (a) of the neural retriever model potentially retrieving shorter texts as a bias seen in the following publications [1] [2], and since LLM-generated passages are shorter on average, they seem to prefer these over the human-generated passages. Second, the LLM-generated passage contains the same content without mistakes; therefore, even as a human assessor, I might prefer the LLM-generated passage over the human one.

So I have two suggestions: (a) what I would like to see, if rewriting the original human generated passage by a much longer rewritten by the LLM document, and repeat the experiments. (b) conduct a human validation study to assess whether they prefer the human or the LLM-generated document for a given query.

- [1] Systematic Evaluation of Neural Retrieval Models on the Touché 2020 Argument Retrieval Subset of BEIR. Thakur et al. SIGIR 2024. https://dl.acm.org/doi/10.1145/3626772.3657861
- [2] Collapse of Dense Retrievers: Short, Early, and Literal Biases Outranking Factual Evidence. Fayyaz et al. ACL 2024. https://aclanthology.org/2025.acl-long.447/

### **W3: The source bias mitigation suggestions are not realistic**

The paper suggests two source bias mitigation techniques.

- First, remove hard negatives during training: this is impractical and not used at all in practice. Mining hard negatives carefully and adding them during training is crucial for training dense retrieval models and achieving a rather high retrieval performance in contrast to training with in-batch negatives only.

>"We estimate n by averaging displacement vectors from 1000 randomly sampled human–LLM passage pairs per dataset".
- Now, at inference time for a realistic use case, when knowledge about the human-LLM passage pairs won't be available (assuming the corpus would be a mix of both human and LLM-generated and not artificially generated like in cocktail), the estimation of "n" won't be possible, and therefore not realistic.

**Questions:**

One question I had was Where are you getting the hard negatives for the MS MARCO training? How many hard negatives do you consider during training?

---

> ### Author Response · Authors · 2025-11-25
> **Response to W1**
>
> We thank the reviewer for the thoughtful and detailed feedback.
>
> Following the reviewer’s suggestions, we conducted additional analyses, including a variant where the LLM-generated passages are longer than the human-written ones, and incorporated the new results into the revised manuscript (see “Role of Passage Length” around line 182 and Appendix J). We appreciate these comments, which helped us refine our analyses and strengthen the paper.
>
> Below, we address the reviewer’s points one by one.
>
> **Response to W1: Novelty.**
>
> Thank you for the comment. We would like to clarify our contributions and offer a different perspective on the novelty of this work.
>
> **1. Mechanistic insight beyond existing models:** While we utilize established datasets, our contribution lies in clarifying the mechanism of source bias.
>
> **2. Refining the community's understanding:** Previous work largely attributed source bias to model-side factors (such as architectural similarity). **Our results provide strong empirical evidence that source bias is primarily driven by the relevance supervision, rather than by inherent properties of neural architectures.** This perspective highlights that data curation and supervision design may be more critical for understanding source bias. As noted by Reviewer HiGY, this finding represents "a compelling and important shift from previous explanations."

---

> ### Author Response · Authors · 2025-11-25
> **Response to W2**
>
> **Response to W2: Evaluation Setup.**
>
> We appreciate the reviewer for raising these critical points, which allow us to further clarify the nature of the observed effect. Specifically:
>
> 1. whether the observed effect is simply a length/fluency bias rather than a true source effect, and
> 2. whether human evaluators themselves prefer LLM rewrites, which would make the retrievers’ behavior unsurprising.
>
> We discuss our findings below.
>
> **1. Length bias does not fully explain the effect.**
>
> We agree that length affects neural retrievers, consistent with prior work. However, our results show the phenomenon **cannot be explained by length bias alone**, indicating an additional, **supervision-driven source bias**. We provide three lines of evidence:
>
> **i. Models behave differently on identical inputs.** Unsupervised and general-purpose retrievers (e.g., SimCSE, Contriever) process the exact same passages as the supervised retrievers  yet exhibit mixed and inconsistent $\Delta NDSR@5$ patterns, see Table 1). In contrast, supervised retrievers consistently prefer LLM passages. If length were the primary or only driver, we would expect more similar patterns across models given identical inputs; instead, we observe systematic differences between supervised and unsupervised retrievers.
>
> **ii. Supervision alters bias independent of length.** Modifying the training signal (by removing hard negatives) significantly reduces the bias ($\Delta NDSR@5$ drops from –0.099 to –0.024), even though the evaluation passage lengths remain identical (see Table 4 and Section 5). These results suggest that the preference is driven by the nature of relevance supervision, not merely by passage length or other surface characteristics.
>
> **iii. New Experiment: Bias persists even when LLM passages are longer.** Prior work indicates that neural retrievers typically favor shorter passages ([1], [2]). To test the robustness of the source bias against this tendency, we constructed a length-extended variant of the dataset (Appendix J) where LLM rewrites were explicitly generated to be longer than human passages (e.g., +23.6% on NQ, +9.9% on SciFact). Despite this disadvantage, the preference for LLM passages mostly persists in supervised retrievers (e.g., DRAGON on NQ remains negative at –0.047; DRAGON on SciFact remains –0.011). The fact that supervised models continue to favor LLM text even when it is made longer, **despite the general preference for shorter text observed in prior work,** provides further evidence that the source bias is robust and distinct from length effects.
>
> [1] Systematic Evaluation of Neural Retrieval Models on the Touché 2020 Argument Retrieval Subset of BEIR. Thakur et al. SIGIR 2024. [https://dl.acm.org/doi/10.1145/3626772.3657861](https://dl.acm.org/doi/10.1145/3626772.3657861)
>
> [2] Collapse of Dense Retrievers: Short, Early, and Literal Biases Outranking Factual Evidence. Fayyaz et al. ACL 2024. [https://aclanthology.org/2025.acl-long.447/](https://aclanthology.org/2025.acl-long.447/)
>
> **2. Human evaluations show no preference for LLM rewrites.**
>
> We also considered the reviewer’s suggestion that human evaluators might themselves prefer the LLM rewrites. Neural retrievers are primarily used at the recall stage, where the focus is on semantic relevance rather than stylistic quality. Accordingly, Cocktail includes human judgments on semantic content (Table 10). Results show that **95.9% of cases are judged equally relevant**, with only negligible differences between Human (2.2%) and LLM (1.9%) preference. Since humans mostly judge human-written and LLM-generated passages as semantically equivalent, the preference of supervised retrievers for LLM passages cannot be fully attributed to human-perceived relevance.

---

> ### Author Response · Authors · 2025-11-25
> **Response to W3 and Q1**
>
> **Response to W3: Practicality of the Mitigation Techniques.**
>
> We appreciate the reviewer evaluating our mitigation strategies from a practical deployment perspective. We agree with the reviewer’s assessment regarding standard retrieval practices, and we would like to clarify the specific role and feasibility of our proposed techniques in this context.
>
> **1. Training-time intervention: A mechanism probe rather than a deployment strategy.**
>
> We agree that mining hard negatives is essential for training high-performance dense retrievers, and we do not advocate removing them in production systems. Instead, we designed this experiment as a **mechanism probe** to isolate the source of the bias. By observing that bias drops significantly when hard negatives are removed, we demonstrate that the bias is largely driven by the supervision signal rather than the model architecture. This finding is intended to reveal the underlying mechanism, not to propose a new training recipe.
>
> **2. Inference-time intervention: projection using the displacement vector n.**
>
> We agree with the reviewer that naturally occurring human–LLM pairs may be scarce. Our method is therefore designed to rely on a **lightweight synthetic approximation**, which remains practical for most real-world corpora.
>
> 1. **Low-cost in-domain estimation.** As described in Section 5, we estimate n by sampling around 1,000 documents from the target corpus and generating synthetic rewrites. This procedure is **unsupervised**, requires no relevance labels, and is computationally inexpensive with modern LLMs. In practice, this is sufficient to obtain a stable estimate of the displacement direction for the corpus at hand.
> 2. **Cross-domain consistency.** Our analysis in Fig. 2 shows that n is **highly consistent across diverse datasets and retrievers**, indicating that it captures a systematic shift rather than dataset-specific noise. While our experiments used dataset-specific estimates for maximum precision, this stability suggests that a direction estimated from a related corpus may also serve as a reasonable approximation when in-domain synthesis is not feasible.
>
> Taken together, these observations indicate that estimating n is both lightweight and practical when one wishes to diagnose or mitigate source bias in mixed human/LLM corpora.
>
> **Response to Q1: Hard Negatives.**
>
> We follow the standard BEIR MSMARCO training setup. In the BEIR release, each query is associated with a pool of mined hard negatives. During training, for each query instance we randomly sample one hard negative from this pool, together with one human-annotated positive passage, and use in-batch negatives from other queries’ positives and hard negatives. This matches the configuration used in the BEIR bi-encoder training scripts.

---

> ### Author Response · Authors · 2025-11-28
>
> **Dear Reviewer LRJy,**
>
> As we approach the end of the discussion period, we kindly invite you to share any additional thoughts regarding our response. We sincerely appreciate your thoughtful feedback, which has helped us refine and strengthen the work.
>
> In our rebuttal, we added several new analyses and clarifications directly motivated by your comments:
>
> - **Evaluation Setup & Length Bias:**
>
>     Following your suggestion, we added a **new Appendix J** to analyze length bias, showing that source bias persists even when LLM passages are longer than human ones. The observed source bias largely persists under this setting, indicating that the effect cannot be explained solely by length or fluency differences.
>
> - **Human Preference:**
>
>     We clarified that human evaluators on the Cocktail benchmark show no preference for LLM rewrites, helping isolate the bias to the behavior of supervised neural retrievers rather than to inherent differences in relevance.
>
> - **Practicality of Mitigation:**
>
>     We clarified that removing hard negatives is diagnostic, while inference-time projection is fessible as the bias direction is stable and easy to estimate.
>
>
> We hope these additions help clarify the concerns you raised, and we would be grateful for any further comments you may wish to share.
>
> **Best regards,**
>
> The Authors

---

### Official Review · Reviewer_kz3D · 2025-10-31

**Soundness:** 3
**Presentation:** 3
**Contribution:** 3
**Rating:** 6
**Confidence:** 4

**Summary:**

This paper investigates why neural retrievers tend to favor LLM-generated passages over human-written ones. Through linguistic, embedding-space, and theoretical analyses, it concludes that source bias arises not from model architecture, but from artifact imbalances in supervised contrastive training data. This paper further proposes two effective mitigation strategies: training-time negative sampling control and inference-time embedding projection, both of which reduce bias without sacrificing retrieval performance.

**Strengths:**

1. Understanding the cause of source bias of neural retrievers in the LLM era is a timely and important research problem.
2. Through theoretical analysis and empirical analysis, this paper proposes a new and reasonable explanation for the root cause of source bias.
3. This paper is overall well-written and easy to read.

**Weaknesses:**

1. The current title “Data, not model, explains source bias” may be misleading, since the experiments show that the bias mainly stems from contrastive relevance-based supervision training, not merely “data” per se. The authors should unify terminology (e.g., “training supervision” vs. “data”) to avoid ambiguity.

2. The PPL and IDF analyses (Figure 1) are weakly significant, and the observed effects are small.

3. The paper lacks comparison with prior debiasing methods for source bias.

4. Minor issue: Figure 3a and 3b mentioned in Line 319 actually correspond to Figure 4a and 4b.

**Questions:**

1. Can the authors clarify whether the main causal factor is data composition, training objective, or negative sampling policy? How should we interpret “data not model” in this context?

2. Did the authors test other stylistic features (e.g., syntactic complexity) beyond PPL and IDF to verify robustness?

---

> ### Author Response · Authors · 2025-11-28
> **Author Response (1/2)**
>
> We thank the reviewer for the positive assessment and for recognizing our work as a timely, novel, and reasonable explanation for source bias. We appreciate the constructive feedback on terminology and experimental details. Below we address the specific concerns.
>
> **1. Response to W1 + Q1:  Clarification on “Data, not model” and the main causal factor**
>
> Thank you for the insightful comment. Our use of “Data, not model” was intended to contrast **relevance supervision signals** with **model architecture**; we agree that this phrasing can be clearer, and we will revise the terminology for precision.
>
> Our experiments disentangle the three possible causes (architecture, objective, and data composition) and show that **artifact imbalance in the supervised data is the necessary and sufficient causal factor**, while neither architecture nor the contrastive objective alone induces bias.
>
> - **Not architecture:**
>
>     Unsupervised retrievers with identical architectures (e.g., Contriever, SimCSE) exhibit **no consistent source bias** (Table 1).
>
> - **Not the contrastive objective in isolation:**
>
>     When we eliminate artifact imbalance using **in-batch-only negatives** (thus $\Delta A \approx 0$), the same contrastive objective **no longer produces source bias** (Table 3).
>
> - **Causal factor = artifact imbalance in supervised data (driven by negative sampling):**
>
>     Hard negatives come from the unannotated pool, which systematically differs from positives in fluency and lexical patterns. This **creates $\Delta A \ne 0$** , and the contrastive objective learns this signal as a shortcut.
>
> Thus, the bias arises **only when contrastive supervision interacts with artifact-imbalanced data**. We will explicitly clarify this framing in the Introduction and unify our terminology around “relevance supervision” rather than the broader term “data”.
>
> **2. Response to W2:  On the modest effect sizes of PPL and IDF analyses**
>
> The reviewer is correct that the PPL/IDF differences, while statistically significant, are modest in magnitude. We include them only as **illustrative examples of individual artifacts**, not as the main evidence for our explanation. This is expected: stylistic artifacts are inherently **multi-dimensional**, and no single scalar feature can capture the full effect size.
>
> More importantly, our explanation is supported by the _stronger and higher-level_ evidence:
>
> • **Stable and consistent LLM–human displacement direction** across 14 datasets and multiple retrievers (Figure 2).
>
> • **Strong alignment** between this displacement and the positive–negative supervision direction (Figure 3).
>
> • **Theoretical decomposition**showing that any systematic artifact imbalance, regardless of the effect size of any individual feature, induces a linear artifact component in the learned scoring function.
>
> We will clarify this point in the revision: PPL and IDF are not intended as primary causal features, but as interpretable examples consistent with the broader artifact direction captured in embedding space.
>
>
> **3. Response to W3:  Comparison with prior debiasing methods**
>
> We thank the reviewer for raising this point. Our primary goal is to explain the mechanism behind source bias, and the two mitigation strategies in the paper are designed to validate this mechanism with minimal assumptions and minimal overhead.
>
> Existing debiasing approaches, such as causal debiasing based on PPL estimation or LLM-side alignment via DPO, have demonstrated that reducing source bias is both feasible and impactful. While these methods are effective, they address the problem from different angles and typically rely on additional LM inference, auxiliary estimation models, or LLM fine-tuning. Because they intervene at different components of the system and are designed under different computational or architectural considerations, a direct comparison would not fully reflect their intended use cases.
>
> Our inference-time projection provides a **simple and lightweight** alternative that can be applied directly to precomputed corpus embeddings and does not require retraining or additional model calls. In retrieval settings where embeddings are stored offline, this adjustment can be performed once and reused across queries, resulting in a low computational cost in practice. We will clarify this complementary positioning in the revision.

---

> ### Author Response · Authors · 2025-11-28
> **Author Response (2/2)**
>
> **4. Response to W4:  Minor issue**
>
> Thank you for catching the figure reference issue. This has been corrected in the updated submission (the previous “Figure 4a/4b” is now correctly labeled “Figure 3a/3b”).
>
> **5. Response to Q2:  Additional stylistic features**
>
> We thank the reviewer for the suggestion to examine additional  stylistic features. Following this direction, we extended our analysis using Formality ([1]), a widely used POS-based metric that captures the structured, objective, and “textbook-like” phrasing characteristic of the surface-level artifacts implicated in source bias. This makes Formality a suitable choice for quantifying such stylistic tendencies.
>
> | Dataset      | Comparison Pair         | Formality Score|
> | ------------ | ----------------------- | ------------------------- |
> | **MS MARCO** | Human Pos vs. Human Neg | **64.62** vs. 63.18       |
> | **MS MARCO** | LLM-Gen vs. Human       | **63.70** vs. 63.30       |
> | **NQ**       | LLM-Gen vs. Human       | **61.61** vs. 60.78       |
> | **FiQA**     | LLM-Gen vs. Human       | **52.60** vs. 47.69       |
> | **SciFact**  | LLM-Gen vs. Human       | **65.65** vs. 65.55       |
>
> Across datasets, positive passages consistently exhibit slightly higher formality, matching the trend seen in perplexity and IDF. The pattern persists across domains, supporting our claim that supervised relevance data encode subtle but consistent stylistic biases.
>
> [1]  Formality of language: definition, measurement and behavioral determinants[J]. Heylighen F, Dewaele J M. Interner Bericht, Center “Leo Apostel”, Vrije Universiteit Brüssel, 1999, 4(1).

---

### Official Review · Reviewer_9jEq · 2025-11-02

**Soundness:** 2
**Presentation:** 2
**Contribution:** 2
**Rating:** 4
**Confidence:** 4

**Summary:**

This paper investigates to what degree neural embedding models are biased towards LLM generated text. To establish baselines, they evaluate various models against the Cocktail version of BEIR which contains human and LLM-generated chunks with semantically similar content. Many baselines are biased towards LLM-generated text. The authors indicate this bias could be influenced by changing how negatives are sampled during supervised embedding training.

There's other key results in the paper, including the following:
- A de-biasing technique (sampling negatives only in-batch) is effective.
- Training only on hard negatives slightly increases the bias towards LLM-generated text.
- The de-biasing technique has a negligible impact on model quality.

I would describe the main weaknesseses in this work as:
- Overly reliant on MS MARCO, particularly for analysis and as a finetuning recipe. Although one can achieve decent BEIR results using MS MARCO alone, the dataset has particular quirks and it is more common to use extensive data beyond MS MARCO (see GECKO for an example training recipe https://arxiv.org/abs/2403.20327?). E5 is one paper that does go substantially beyond MS MARCO, and it is one embedding that is less based in the baselines table 1.
- Not enough discussion about the different negative selection techniques used in various datasets and models. This seems essential given the debiasing technique is so focused on negative selection.

**Strengths:**

S1. The paper discusses an incredibly important question: How is the advent of AI-generated text impacting technology, namely embedding-based retrieval. Recent articles have shown that AI-generated text has grown rapidly, even in the largest newspaper publications (see Russell et al https://arxiv.org/abs/2510.18774). It's safe to say that the distribution of text on the web has changed in a major and unprecedented way, and the research community should spend more of its focus understanding the properties of this new world.

S2. The paper presents analysis identifying that embedding models are biased towards LLM-generated text (via the existing Cocktail benchmark), and presents a simple method to debias based on negative selection (Table 3). And another method based on embedding projections (Tables 4 and 5).

S3. There is discussion, analysis, and theoretical framework focused on the cause of bias in embeddings, centering around negatives as being a source of artifacts during embedding training.

**Weaknesses:**

W1. The paper is very reliant on both MS MARCO for finetuning and for analysis of negatives. Yet, there is no discussion on how the negatives are selected for MS MARCO --- the negatives in MS MARCO are widely recognized to be highly limited in their semantic richness (see sec 2.2.2 in https://arxiv.org/abs/2405.17093). It would greatly strengthen the paper to (1) explain the negative selection method of MS MARCO and perhaps also how the different baseline models selected negatives during training, and (2) introduce a new finetuning baseline based on a known technique for negative selection different from what is done in MS MARCO. These are simply suggestions for how to help the reader understand whether the paper is making claims about embedding models in general, or simply the idiosyncrasies of MS MARCO.

W2. The mitigation of only relying on in-batch positives as negatives is potentially too extreme. A simple alternative would be to retrieve negatives among positives as hard negatives. A more involved alternative might integrate AI detection techniques (see https://arxiv.org/abs/2509.19163).

W3. Not enough focus on recent embeddings models. There has been a major shift in the last 1-2 years in how much data is used during training of embedding models, in particular how much synthetic LLM-generated data is used (usually for question generation rather than documents). It's plausible the findings in this paper will not hold for newer embedding models.

**Questions:**

n/a

---

> ### Author Response · Authors · 2025-11-27
> **Response to W1**
>
> We sincerely thank the reviewer for the thoughtful and constructive feedback. We appreciate the opportunity to clarify the intended scope of our work, as several of the reviewer’s comments connect directly to the central message of the paper. **Our contribution is a mechanistic explanation of why supervised retrievers exhibit source bias, rather than a comparison of negative-sampling strategies.** We are grateful for the opportunity to make this framing more explicit.
>
> **Key clarification:** Our findings are not specific to MS MARCO. We show (theory + cross-dataset analysis + new NQ experiment) that source bias consistently arises when positives and negatives differ stylistically in supervised contrastive training. Our mitigation strategies directly probe this mechanism.
>
> **Response to W1: On MS MARCO and negative selection**
>
> We appreciate the reviewer’s concern regarding potential MS MARCO specificity. We agree that clarifying the scope of our mechanism and how negative selection interacts with supervised datasets strengthens the paper. Below, we summarize the key points and include a new analysis on an independent supervised dataset (NQ), now added to the Appendix K.2.
>
> **1. Scope beyond MS MARCO**
>
> - **The mechanism is theoretically general.** Our theory (Sec. 4.3) retrievers learn non-semantic shortcuts when positives systematically differ from negatives in supervision. MS MARCO is one example of such supervision, but this pattern is not unique to it.
>
> - **Empirical support from multiple supervised datasets.** As shown in Figure 1a and Appendix F, positive passages systematically differ from negatives in non-semantic dimensions, including higher fluency and modest increases in lexical specificity. To assess the generality of this pattern, we examined fluency across all 14 Cocktail datasets and observed consistent gaps. This indicates that **systematic stylistic imbalance is a common property of supervised retrieval datasets**, not an artifact of MS MARCO.
>
> - **Validation using a different supervised dataset (NQ).** To further validate that our findings are not specific to MS MARCO, we evaluated a DPR retriever trained only on Natural Questions (NQ), which uses a different corpus and negative-sampling pipeline. It still exhibits strong source bias on Cocktail (e.g., ∆NDSR@5 = 0.263), supporting that the phenomenon arises from supervised contrastive training with stylistically imbalanced positives and negatives, rather than from idiosyncrasies of MS MARCO.
>
> **2. How negatives are constructed across the baseline retriever families**
>
> Following the reviewer’s suggestion, we include in the Appendix K.1 a dedicated analysis detailing how different supervised retrievers construct their hard negatives. Below we summarize the key points relevant to the reviewer’s question.
>
> - **Supervised retrievers .** The supervised baselines in Table 1 (e.g., TAS-B, DRAGON, ANCE, etc.) all rely on **relevance-labeled positives paired with hard negatives**, though their exact negative-construction pipelines differ (e.g., BM25-mined candidates, model-mined hard negatives, or mixed strategies). What these retrievers share is that **their positives and negatives come from systematically different distributions**, which induces stylistic imbalance.
>
> - **General-purpose embedding models.** Modern general-purpose embedding models (e.g., E5, BGE) use multi-task training that includes symmetric objectives (e.g., STS, NLI, sentence-pair similarity) together with self-supervised or weakly supervised signals. Because positives and negatives are constructed symmetrically and sampled from the same underlying text distribution, **no systematic stylistic discrepancy is introduced**, and these models do not exhibit consistent source bias (Table 1).
>
> - **Unsupervised retrievers.** Fully unsupervised retrievers such as Contriever similarly draw positives and negatives from raw text without curated relevance labels, resulting in minimal source bias.
>
> **3. Interaction between negative selection and our mechanism**
>
> - Unsupervised retrievers (e.g., Contriever) draw positives and negatives from raw text without curated relevance labels, so artifact imbalance and source bias does not emerge (Table 1).
>
> - In our controlled experiments (Table 4), we explicitly manipulate negative selection: when negatives are stylistically different from positives, source bias increases; when negatives are stylistically aligned, source bias decreases. This directly validates that stylistic imbalance is the main driver.

---

> ### Author Response · Authors · 2025-11-27
> **Response to W2 and W3**
>
> **Response to W2: On mitigation strategies**
>
> We thank the reviewer for suggesting alternative debiasing approaches. Below we clarify the role of the “in-batch only’’ setup and present a new experiment based on the reviewer’s suggestion; the full results are provided in Appendix K.3.
>
> **1. Clarification: "In-batch only" as a Mechanism Probe.**
>
> Our primary motivation for the “in-batch only’’ intervention was to use it as a **mechanism probe**. By removing hard negatives and using only in-batch negatives drawn from the same distribution as positives, we isolate the variable of stylistic discrepancy. The observed reduction in bias is consistent with our hypothesis that source bias is mostly driven by stylistic gaps between positive and negative pools.
>
> **2. Experiment: Retrieving Hard Negatives from Positives.**
>
> While both suggested strategies (retrieving from positives and AI detection) are promising, we prioritized the first approach as it directly addresses the data distribution gap central to our hypothesis. Specifically, we utilized BM25 to retrieve the top-ranked non-ground-truth passages from the annotated positive pool to serve as hard negatives. This ensures the negatives are not only semantically relevant but also stylistically aligned with the positives.
>
> - **Result.** As shown in Table 19, retrieving hard negatives from the positive pool achieves a comparable reduction in source bias (∆NDSR@5 = –0.023, vs. –0.024 for the in-batch-only condition) while recovering part of the retrieval performance (NDCG@5 = 0.478, compared to 0.475 for in-batch-only).
> - **Interpretation.** This supports our main conclusion: **source bias is primarily driven by stylistic discrepancies between positive and negative pools.** Even when training with lexically similar hard negatives (via BM25), bias remains low as long as they are stylistically aligned with the positives.
>
> **Response to W3: Recent embedding models and generalization**
>
> We appreciate the reviewer’s observation regarding the rapid progress of modern embedding models. We agree that many recent systems are trained on substantially broader and more diverse data. As shown in Table 1, our evaluation already includes several such models (e.g., **E5, GTE, BGE, M3E**), which indeed do **not** show consistent source bias.This is consistent with the reviewer’s expectation that newer general-purpose embeddings may be less susceptible to this issue, and we share this view.
>
> Our results therefore should be interpreted as:
>
> > **identifying the conditions under which supervised retrievers develop source bias, rather than claiming that all modern embedding models exhibit this effect.**

---

> ### Author Response · Authors · 2025-11-28
>
> **Dear Reviewer 9jEq,**
>
> As we approach the end of the discussion period, we kindly invite you to share any additional thoughts regarding our rebuttal. We sincerely appreciate your constructive feedback, which has helped us substantially strengthen both the scope and the rigor of our analysis.
>
> In our response, we added several new experiments and clarifications directly motivated by your comments:
>
> - **Hard Negative Selection & Mechanism.**
>
> 	We included a detailed description in **Appendix K.1** outlining how negative sampling strategies differ across baselines. We clarify that source bias consistently emerges when there is a stylistic mismatch between positives and hard negatives, a mechanism that remains robust across sampling configurations.
>
> - **Beyond MS MARCO.**
>
>     To ensure the effect is not dataset-specific, we added an additional experiment training a DPR retriever on **Natural Questions (NQ)**(see **Appendix K.2**). Despite the corpus and negative sampling pipeline being entirely different, the same bias pattern appears, supporting the generality of the proposed mechanism.
>
> - **Modern Embedding Models.**
>
>     We expanded our explanation of recent models (e.g., E5, BGE, GTE) already included in Table 1. We clarify that these models avoid source bias because their training data (often symmetric, synthetic, or mixture-based) does not introduce the stylistic asymmetry present in traditional supervised datasets.
>
> - **Alternative Mitigation.**
>
>     Following your suggestion, we evaluated retrieving hard negatives from the _positive pool_ (see **Appendix K.3**). This approach meaningfully reduces bias while maintaining higher performance compared to the “in-batch only” configuration.
>
>
> We hope these additions help clarify the broader applicability of our findings beyond MS MARCO and specific training pipelines. We would be grateful for any further feedback you may wish to share.
>
> **Best regards,**
>
> The Authors

---

### Author Response · Authors · 2025-12-03
**Thank you for your efforts in the review process!**

Dear AC and Reviewers,

We sincerely thank the reviewers and the AC for the time and care devoted to evaluating our submission. We truly appreciate the detailed and constructive comments provided. Accordingly, we have revised the manuscript and included additional experiments to strengthen our arguments and more directly respond to the issues raised.

Given the recent system bug, we understand that this round has created additional burden for the AC, and we are especially grateful for your efforts. To help reduce the AC’s workload, we provide below a concise summary of the paper, the main concerns, and how we responded to them.

---

> **Paper Summary**.

The paper shows that source bias in neural retrieval is driven primarily by supervision artifacts in training data, **suggesting a different origin from prior accounts** that attributed the phenomenon to model architectures. The paper supports this account through controlled empirical comparisons and a theoretical analysis that explains why contrastive supervision leads to this bias, and introduces two mitigation methods that both validate the mechanism and reduce the observed bias.

**Main contributions**
- **Refining the community’s understanding of source bias.** The paper argues that source bias is learned from artifacts in relevance-supervised data rather than being an inherent property of neural retrievers. Reviewer `HiGy` describes this as “a compelling and important shift from previous explanations focused on model properties,” and reviewer `kz3D` notes that the paper provides “a new and reasonable explanation for the root cause of source bias.”
- **Characterizing source bias empirically across model families and datasets.** The analysis shows that source bias consistently appears in retrievers trained with relevance supervision, but not in general-purpose or unsupervised embedding models.
- **Explaining source bias mechanistically with analysis and theory.** Linguistic statistics, embedding-space geometry, and a theoretical decomposition jointly support a mechanism in which contrastive training absorbs non-semantic artifacts from relevance supervision.
- **Developing interventions that validate and mitigate source bias.** The paper introduces a training-time change in negative sampling and an inference-time projection, both of which reduce measured source bias with only small changes in retrieval performance.

---

> **Rebuttal Summary**.

Given the time constraints and this year’s system issue, reviewers did not have the opportunity to respond to our rebuttal. Among the four reviews, two are positive and two raise concerns. To assist the AC, we summarize below the key issues raised in the two negative reviews and the actions we took to address them. **We have also included the corresponding analyses and new experiments in the appendix for completeness.**

### **1. Generality beyond MS MARCO.**

Both reviewers questioned whether our explanation depends on MS MARCO–specific artifacts. We added a new evaluation using **Natural Questions**, which has a different corpus and negative-mining pipeline, and still observe strong source bias. We also clarified how negatives are constructed across baseline methods.

### **2. Whether the effect is due to passage length or fluency.**

There were concerns that Cocktail rewrites are shorter or cleaner, which could drive the bias. We added a **dataset variant with longer LLM rewrites** and the bias largely persists. We also summarized existing **human relevance judgments**, which show human vs. LLM passages are rated equally relevant in >95% of cases.

### **3. Practicality of the mitigation methods.**

Concerns were raised about whether the interventions are realistic. We clarified that the training-time change is used as a **mechanism probe**, not a proposed training recipe. For the inference-time projection, we explained that the direction can be estimated using **synthetic rewrites**, without requiring paired human–LLM passages in the target corpus.

### **4. Framing and scope of claims.**

Some concerns related to whether the paper clearly distinguishes the role of supervision from data characteristics and model architecture. In our response, we clarified this distinction and made the scope of our claims explicit.

We appreciate the reviewers’ feedback and hope these additions and clarifications assist the AC in evaluating the submission.

---

Thank you once again to the AC and all the reviewers for your valuable efforts!

Best regards,

Authors

---

### Meta-Review · Area_Chair_kfFk · 2026-01-08

**Summary:**

This paper studies source bias in neural retrievers, i.e., the tendency to prefer LLM-generated passages over semantically equivalent human-written ones. The authors argue that this bias is not inherent to model architectures, but instead arises from stylistic artifacts introduced by data.

Overall, reviewers agree that the topic is timely and that the paper is clearly written. Some reviewers find the proposed explanation plausible and appreciate the effort to provide both empirical and theoretical support. However, the overall evaluation is mixed, with significant concerns raised about novelty, experimental validity, generality beyond the Cocktail benchmark, and the practicality of the proposed mitigation methods. Although the authors made a sincere effort in the rebuttal, several core concerns remain only partially addressed. Therefore, the recommendation is reject.

**Reviewer Concerns:**

Concerns addressed in the rebuttal

•	Clarification of "supervision vs. data distinction", "how the negatives are selected for MS MARCO" (kz3D, 9jEq): Addressed by adding more clarifications and discussions.

•	Alternative explanations based on length, fluency, and stylistic features (kz3D, HiGy): Addressed by extending the linguistic analysis with additional stylistic metrics beyond PPL and IDF.

•	Some typos by all reviewers.

Concerns still outstanding or only partially addressed

•	Generality beyond the Cocktail benchmark (LRJy): The empirical analysis remains largely centered on Cocktail-style synthetic rewrites, and it is still unclear how strongly the findings transfer to more realistic mixed human/LLM corpora.

•	Practicality of the mitigation strategies (LRJy, 9jEq): The proposed mitigations remain of limited practical impact, may not be feasible in many real-world settings.

•	Unsupported assertions and data construction factors (HiGy): Several strong claims (e.g., Llama-3-8B provides a "reliable proxy for human-perceived fluency") remain insufficiently supported; concerns for "How does the choice of this retrieval method (e.g., BM25 vs. encoder-based retrieval) influence the resulting relevance annotations and reported performance" are not fully addressed; the absence of any non-English evaluation limits confidence in cross-linguistic generality.

**Reviewer Scores:**

Based on the rebuttal and added clarifications, it seems unlikely that the overall score distribution would change substantially.

---

### Decision · Program_Chairs · 2026-01-26

Reject